# Nuclear receptor NR2F6 inhibition potentiates responses to PD-L1/PD-1 cancer immune checkpoint blockade

Victoria Klepsch [1], Natascha Hermann-Kleiter[1], Patricia Do-Dinh[1], Bojana Jakic[1], Anne Offermann[2], Mirjana Efremova[3], Sieghart Sopper [4], Dietmar Rieder[3], Anne Krogsdam[3], Gabriele Gamerith[4], Sven Perner[2], Alexandar Tzankov [5], Zlatko Trajanoski [4], Dominik Wolf[4,6] & Gottfried Baier [1]

Analyzing mouse tumor models in vivo, human T cells ex vivo, and human lung cancer samples, we provide direct evidence that NR2F6 acts as an immune checkpoint. Genetic ablation of *Nr2f6*, particularly in combination with established cancer immune checkpoint blockade, efficiently delays tumor progression and improves survival in experimental mouse models. The target genes deregulated in intratumoral T lymphocytes upon genetic ablation of *Nr2f6* alone or together with PD-L1 blockade reveal multiple advantageous transcriptional alterations. Acute *Nr2f6* silencing in both mouse and human T cells induces hyper-responsiveness that establishes a non-redundant T-cell-inhibitory function of NR2F6. NR2F6 protein expression in T-cell-infiltrating human NSCLC is upregulated in 54% of the cases (*n* = 303) and significantly correlates with PD-1 and CTLA-4 expression. Our data define NR2F6 as an intracellular immune checkpoint that suppresses adaptive anti-cancer immune responses and set the stage for clinical validation of targeting NR2F6 for next-generation immuno-oncological regimens.

[1] Division of Translational Cell Genetics, Medical University of Innsbruck, 6020 Innsbruck, Austria. [2] Pathology of the University Hospital Schleswig-Holstein, Campus Luebeck and Research Center Borstel, Leibniz Lung Center, 23538 Leubeck, Germany. [3] Biocenter, Division of Bioinformatics, Medical University of Innsbruck, 6030 Innsbruck, Austria. [4] Tumor Immunology, Tyrolean Cancer Institute & Internal Medicine V, 6020 Innsbruck, Austria. [5] Department of Pathology, University of Basel,  University Hospital Basel, 4031 Basel, Switzerland. [6] Medical Clinic III, Oncology, Hematology & Rheumatology, University Clinic Bonn, 53127 Bonn, Germany. Correspondence and requests for materials should be addressed to G.B. (email: gottfried.baier@i-med.ac.at)

Dysfunction of appropriate immune responses is a crucial event in cancer development and progression. The development of immuno-oncological therapies is likely to revolutionize cancer care in the future[1–10]. Particularly, antibodies blocking cell surface immune system inhibitory receptors, such as cytotoxic T lymphocyte-associated protein 4 (CTLA-4) and programmed cell death-1 and its ligand-1 (PD-1)/(PD-L1), referred to as "immune checkpoints", have been approved for many cancer types including melanoma, non-small cell lung cancer (NSCLC), urothelial cancer and Hodgkin's Lymphoma[11–17]. In addition, adoptive T-cell transfer and cancer vaccines are also providing more encouraging results, especially when combined with antibodies antagonizing the above-mentioned immune checkpoints[18–20]. Even though encouraging, the percentage of patients profiting from these therapies is limited, highlighting the need to further improve immune-activating therapy concepts. Particularly, the identification and validation of alternative and potentially additive or even synergistic immune checkpoint candidates is envisioned to improve immunotherapies and extend their benefits to a larger number of cancer patients.

Pursuing this vision, we considered it reasonable to explore the checkpoints that might be located inside the immune cells, besides those present on the cell surface, and might be suitable targets for future cancer drugs. Following this line of argumentation, we demonstrated the crucial role of the lymphocyte-intrinsic orphan nuclear receptor NR2F6 (Nuclear receptor subfamily 2, group F, member 6; alias Ear2 and COUP-TFIII) as an intracellular checkpoint candidate fine-tuning adaptive immunity. NR2F6 induced an anti-inflammatory signal in the T-cell compartment. Consistent with this observation, Nr2f6-deficient mice spontaneously developed a late-onset autoimmune-type phenotype and were hypersusceptible to induction of experimental neuroinflammation[21]. Mechanistically, NR2F6 acts as an essential signaling intermediate and sets the threshold of T-cell effector functions by acting as a transcriptional repressor that directly antagonizes the DNA accessibility of activation-induced NFAT/AP-1 transcription factors at key cytokine gene loci such as Il2 and Ifng[22].

Recently, it has been shown that the genetic elimination of NR2F6 improves intratumoral CD4+ and CD8+ T-cell infiltration as well as effector functions at the tumor site in transplantable B16-OVA mouse tumor model systems[23]. Importantly, these tumor growth effects depend primarily on NR2F6 function in both CD4+ and CD8+ effector (but not regulatory) T-cell subsets. However, a rigorous assessment of the physiological relevance of NR2F6 function in clinically relevant mouse cancer model systems as well as in human T-cell biology has not yet been carried out. Employing methylcholanthrene (MCA)-induced sarcoma as well as MC38 colon adenocarcinoma model systems and human NR2F6 knockdown T-cell cultures as well as analyzing tumor-infiltrating lymphocytes (TIL) in human NSCLC biopsy samples, we here provide strong pre-clinical evidence that upregulation of NR2F6 at the tumor site renders effector T cells incapable of mounting sufficient anti-cancer immune response. Most importantly, combined genetic ablation of NR2F6 with the established PD-L1 checkpoint blockade is strongly synergistic. Furthermore, these clear anti-tumor immune responses in the Nr2f6−/− therapy groups were not accompanied by any signs of immune-related adverse events (irAE). Thus, our study sets the stage for future clinical validation of NR2F6 targeting as a mechanistically independent and potentially synergistic option to improve the efficacy of immuno-oncology therapies.

## Results

**Loss of NR2F6 function augments tumor rejection**. Nr2f6-deficient mice were significantly more resistant to development of

MCA-induced sarcoma than wild-type mice (Fig. 1a) and exhibited reduced tumor growth during challenge with a primary sarcoma cell line derived from MCA-treated wild-type mice (Fig. 1b, c). Intratumoral immune cells in the MCA sarcoma model contained more abundant CD45+ and CD3+ lymphocytes in Nr2f6−/− versus wild-type Nr2f6+/+ mice (Fig. 1d, e). The CD3+ pool of TILs from these tumors showed no preference for the CD4+ or the CD8+ population (Supplementary Fig. 1H, I). In addition to the spontaneous MCA tumor model, we challenged wild-type and Nr2f6-deficient mice subcutaneously with the transplantable colon adenocarcinoma cell line MC38 and observed decreased tumor outgrowth (Fig. 1f) and a significant survival benefit (Fig. 1g) in Nr2f6−/− mice. These results firmly establish that the genetic ablation of Nr2f6 improves immune-mediated tumor control, finally resulting in a striking benefit in these advanced mouse models relevant to clinical cancer.

Of note, the augmented anti-tumor immune response in Nr2f6−/− mice were linked to a markedly higher PD-1 and PD-L1 expression on CD8+ and CD4+ T-cell subsets in vitro when compared to wild-type animals (Supplementary Fig. 1A–D), suggesting a potential resistance mechanism via upregulation of T-cell exhaustion markers. Interestingly, PD-L1 expression was increased not only in CD45+ leukocytes (Supplementary Fig. 1F) but also in CD45− cells (Supplementary Fig. 1E) when comparing wild-type and Nr2f6-deficient B16-OVA tumor-bearing mice on d14 after tumor challenge, suggesting that loss of NR2F6 in T cells overcomes even high expression of these well-known immune-suppressive checkpoint receptors. In order to exclude "emergency" granulo-monocytopoesis as a main or contributing factor to the enhanced tumor rejection in Nr2f6-deficient mice, hematopoietic stem cells, myeloid progenitor cells and myeloid cells within the bone marrow were investigated following tumor induction. In line with this observation also no differences between wild-type and Nr2f6-deficient myeloid cells such as neutrophils (CD45+Ly-6G+CD11b+), eosinophils (CD45+Ly-6G−SSChi) or monocytes (CD45+Ly-6G−CD11b+), within latter population divided into CD115+Ly-6Clo and CD115+Ly-6Chi cells, were observed (Supplementary Fig. 1J, K). Together, in Nr2f6-deficient tumor-bearing mice, no significant differences in the frequency or number in either of these innate immune cell populations could be detected on day 14.

**Genetic ablation of NR2F6 augments PD-L1 blockade therapy**. Based on these findings, we next studied immunotherapy regimens targeting different checkpoints concomitantly. Challenging Nr2f6+/+ and Nr2f6−/− mice with a high tumor load of $5 \times 10^5$ B16-OVA or $7.5 \times 10^5$ MC38 colon carcinoma cells, all wild-type mice receiving control IgG injections rapidly developed tumors and had to be sacrificed due to ethical reasons at the latest by d21 post injection. In line with our previous results, tumor outgrowth in IgG2b-treated control group of Nr2f6−/− mice was significantly delayed even with this high tumor load and was well comparable to the benefit of blocking the interaction of PD-1 with its ligand PD-L1 in vivo in wild-type mice (Fig. 2a–d). As a remarkable result, however, a clear synergistic benefit of PD-L1 blocking in Nr2f6−/− mice could be observed. This combination therapy was clearly more potent than targeting a single immune checkpoint and even led to rejection of tumors upon challenge with such very high tumor cell numbers of both the B16-OVA (Fig. 2a, b) as well as the MC38 mouse tumor cell line (Fig. 2c, d and Supplementary Fig. 2A−E). Of note, immune infiltration analyses in vivo of the Nr2f6 knockout group in those high-dose tumors models, in principle, recapitulated the situation of the low dose model (see Supplementary Fig. 3A, B). This provides strong preclinical evidence that NR2F6 and PD-1 signaling may act

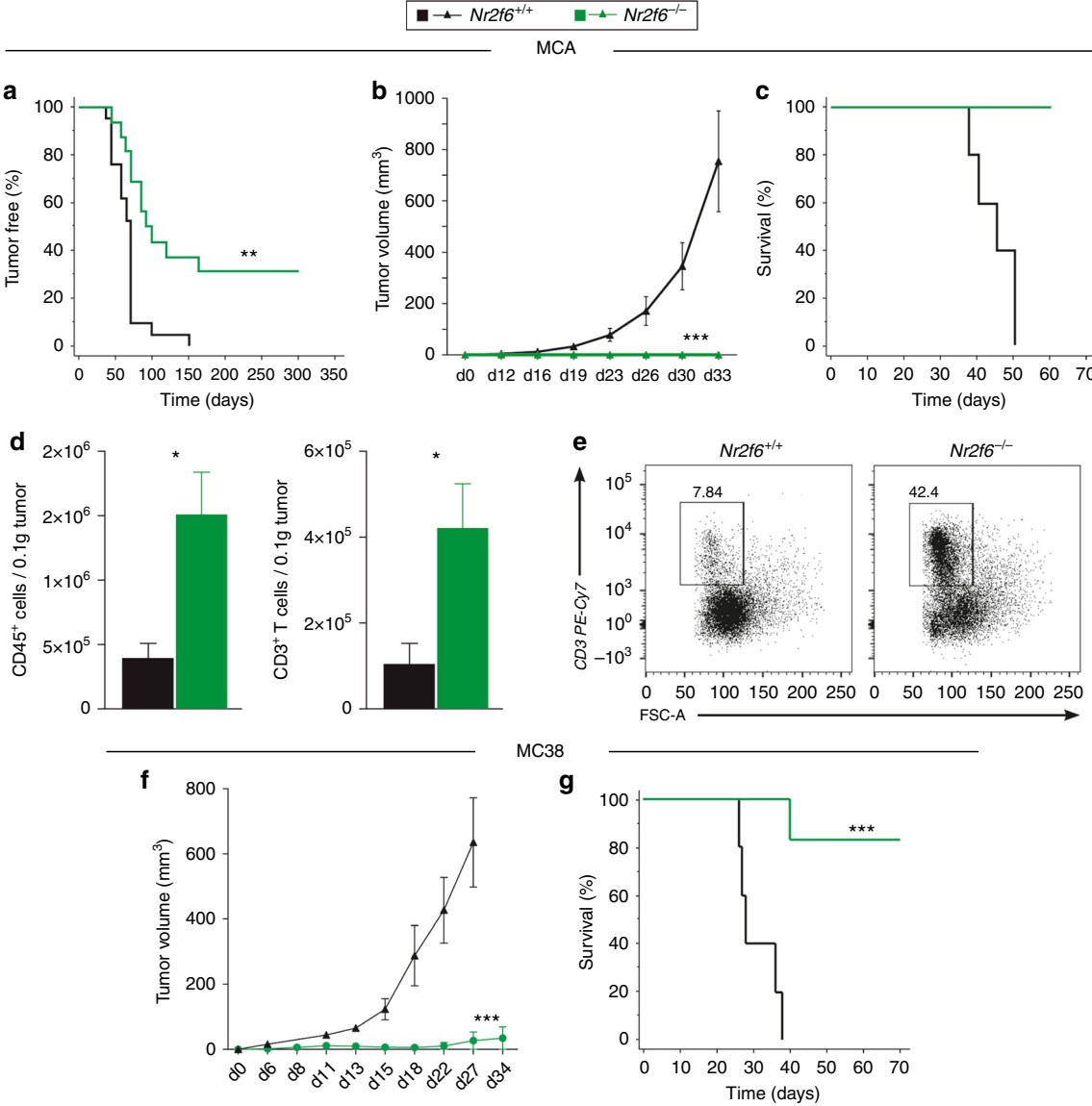

**Fig. 1** $Nr2f6^{-/-}$ mice reject transplantable chemically and induced subcutaneous tumors. **a** Wild-type ($n = 15$) and $Nr2f6$-deficient ($n = 18$) mice were injected with 200 µg MCA and monitored over 300 days; the percentage of sarcoma-free mice is shown ($p = 0.0007$, ANOVA). From the MCA-induced pool of mice, several cell lines were generated. Tumor growth (**b**) and survival (**c**) of one representative wild-type tumor cell line re-injected into wild-type and $Nr2f6^{-/-}$ mice ($n = 6$) with $1 \times 10^6$ cells/mouse is shown ($p < 0.0001$, **b** ANOVA, **c** log-rank test). **d** Significantly enhanced absolute cell numbers of tumor-infiltrating CD45+ leukocytes ($p = 0.018$, unpaired $t$-test, $n = 5$) and CD3+ T cells ($p = 0.031$, unpaired $t$-test, $n = 5$) per 0.1 g of tumor tissue in wild-type (black) and $Nr2f6^{-/-}$ (green) mice spontaneously developing chemically induced (MCA) sarcomas. **e** Representative dot blots of tumor-infiltrating CD3+ T cells from $Nr2f6^{+/+}$ or $Nr2f6^{-/-}$ mice are depicted. Numbers adjacent to outlined areas indicate the percentage of positive cells relative to parental CD45+ gate. Summary of 2−3 independent experiments is shown and data are expressed as the mean ± SEM. **f** The kinetics of tumor cell growth in $Nr2f6^{+/+}$ ($n = 12$, black) and $Nr2f6^{-/-}$ ($n = 6$, green) mice subcutaneously injected with $5 \times 10^4$ MC38 colon carcinoma cells ($p < 0.0001$, ANOVA) as well as **g** significant survival benefit in $Nr2f6^{-/-}$ by a Kaplan-Meier curve are shown, and statistically analyzed by a log rank test ($p = 0.0007$, log-rank test)

together as "threshold regulators" in host-protective tumor immunity. Despite the dramatically improved clinical outcome in $Nr2f6^{-/-}$ tumor-bearing mice subjected to PD-L1 blocking (combinatorial NR2F6/PD-L1 inhibition group) when directly compared to wild-type mice under mono-therapy, no exacerbated signs of irAE were observed during a follow-up period of 3 months (Table 1). We did not see any significant differences in immune cell infiltrates, colon length or weight gain after anti-PD-L1 treatment in $Nr2f6$-deficient mice, when compared to wild-type mice similarly treated (unpaired Student's $t$-test). To evaluate the contribution of CD4+ and CD8+ T cells to the observed decreased tumor outgrowth, we challenged $Nr2f6^{-/-}$ subcutaneously with $1 \times 10^5$ B16-OVA melanoma cells and treated them with CD4+ and CD8+ depleting antibodies. We could show a significantly enhanced tumor outgrowth in $Nr2f6^{-/-}$ mice treated with a combination of CD4+ and CD8+ depleting antibodies when compared to the IgG2b-treated $Nr2f6^{-/-}$ control group (Supplementary Fig. 2F).

Taken together, genetic NR2F6 ablation acts as a "sensitizer" that allows improved therapeutic activity of the clinically approved PD-1/PD-L1 axis blockade in experimental mouse tumor model systems. Hence, releasing the inhibition imposed by NR2F6 appears to be a suitable strategy to vastly enhance efficacy of cancer immunotherapy.

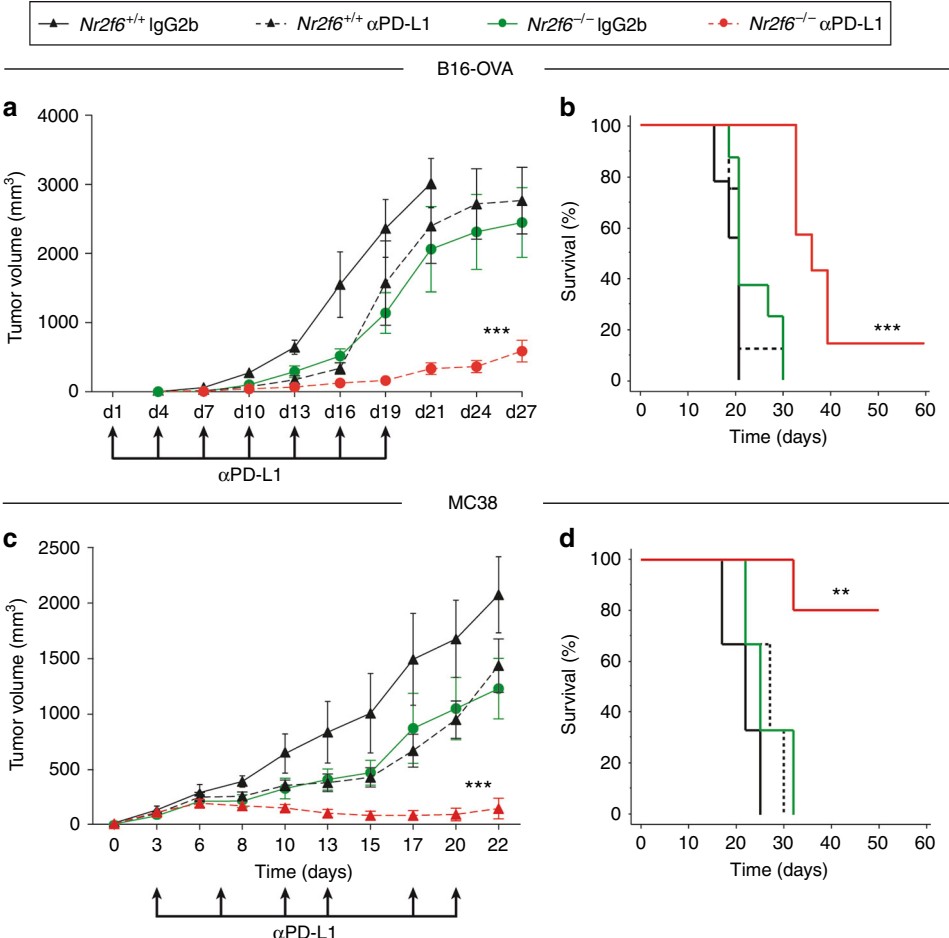

**Fig. 2** Gene ablation of NR2F6 acts as "sensitizer" for the established immune checkpoint blockade in mouse tumor models in vivo. **a** Tumor growth curve ($p<0.0001$, ANOVA, $n = 9$) and **b** Kaplan–Meier survival curve ($p<0.0001$, log-rank test, $n = 9$) of $Nr2f6^{+/+}$ and $Nr2f6^{-/-}$ mice that received the high dose of $5\times10^5$ B16-OVA tumor cells subcutaneously and were treated either with "mono-therapies" of genetic $Nr2f6$ inhibition (green, IgG2b) or PD-L1 blockade in wild-type mice (dashed black, employing the established Ab10F.9G2) or treated with a "combination therapy" (red) ($n = 8$). **c** Tumor growth ($p < 0.0001$, ANOVA) and **d** Kaplan–Meier survival curves ($p = 0.0031$, log-rank test) of $Nr2f6^{+/+}$ and $Nr2f6^{-/-}$ mice that received the high dose of $7.5\times10^5$ MC38 tumor cells subcutaneously and were treated either with "mono-therapies" of genetic $Nr2f6$ inhibition (green, IgG2b isotype control, $n = 7$) or PD-L1 blockade in wild-type mice (dashed black, $n = 10$, the latter employing the established protocol with neutralizing Ab10F.9G2) or with a "combination therapy" (red, $n = 12$) (wild-type IgG2b control group, black, $n = 5$). Results shown are derived from at least two independent experiments. Error bars represent the mean ± SEM

**Coinhibition of NR2F6 and PD-1 alters gene signatures of TIL.** It has been shown that NR2F6-mediated transcriptional repression of key cytokine gene loci such as *Ifng* and *Il2* contributes to an immune suppressed state of tumor antigen-specific effector T cells at the tumor site[23]. However, the specific target genes of NR2F6 on a systemic level remained undefined. It was thus mandatory to further investigate the network of critical target genes suppressed and/or activated by *Nr2f6* gene induction within the tumor microenvironment (TME).

In order to determine the transcriptional signatures of the observed superior cancer immune response associated with genetic *Nr2f6* inhibition, alone and particularly in combination with the established PD-1/PD-L1 axis blocking, we next examined the network of critical target genes in CD3$^+$ TILs, employing a stratified CD45$^+$/CD3$^+$ sorting strategy. As expected, tumor growth in $Nr2f6^{-/-}$ mice was significantly reduced (Supplementary Fig. 3A) and tumors exhibited significantly increased numbers of tumor-infiltrating CD45$^+$CD3$^+$ T cells when compared to wild-type animals, calculated at the level of total numbers on a tumor size basis (Supplementary Fig. 3B). RNA sequencing (RNAseq) analysis of TILs derived from wild-type

and $Nr2f6^{-/-}$ tumor-bearing mice collected at d14 after B16-OVA tumor cell injection clearly demonstrated that the NR2F6- and PD-1-mediated immune-suppressive pathways are mechanistically distinct (Fig. 3a). Further, the control IgG2b-treated or PD-L1 mono-therapy groups were distinctly different with respect to gene expression programs (Supplementary Fig. 3D–F). Simultaneous genetic ablation of NR2F6 and PD-1/PD-L1 interaction blockade significantly affected multiple gene ontology terms analyzed by ClueGO (Supplementary Table 1–6). Detailed analysis of differentially expressed genes with respect to over-represented GO terms showed upregulation of several immune processes related to adaptive immune activation responses such as cell-mediated cytotoxicity, IFNγ signaling, T-cell receptor signaling and TNF signaling pathways (Fig. 3b, c and Supplementary Table 1–6). Processes related to nucleosome assembly or RNA transport were downregulated (Fig. 3d). Furthermore, gene set enrichment analysis showed enrichment of genes associated with T cells in general and Th1 cells in particular (Fig. 3e) in $Nr2f6^{-/-}$ tumor-bearing mice treated with the anti-PD-L1. Several gene signatures known to be favorable for T-cell-mediated tumor rejection (Fig. 3f) were also found to be

**Table 1 Immune-related adverse events (irAE) of MC38-tumor-bearing mouse treatment groups**

| | $Nr2f6^{+/+}$ IgG | $Nr2f6^{-/-}$ IgG | p value | $Nr2f6^{+/+}$ αPD-L1 | $Nr2f6^{-/-}$ αPD-L1 | p value |
|---|---|---|---|---|---|---|
| Colon length (cm) | 10.3 ± 0.1 | 9.4 ± 0.4 | 0.066 | 9.4 ± 0.3 | 9.3 ± 0.3 | 0.82 |
| Spleen weight (mg) | 119 ± 15.2 | 110 ± 17.3 | 0.70 | 112 ± 4.6 | 119 ± 15.9 | 0.69 |
| Spleen cellularity (×$10^7$) | 5.9 ± 0.5 | 4.3 ± 0.7 | 0.11 | 5.0 ± 0.7 | 5.6 ± 0.7 | 0.56 |
| Weight gain d90 (% BW) | 19 ± 3.3 | 20.5 ± 6.5 | 0.89 | 22 ± 2 | 30 ± 5 | 0.42 |
| $CD3^+$ | 29.7 ± 4.6 | 29.9 ± 4.4 | 0.98 | 29.9 ± 1.8 | 30.3 ± 4.1 | 0.94 |
| $CD3^+CD69^+$ | 12.3 ± 1.3 | 13.9 ± 2.7 | 0.62 | 9.5 ± 1.5 | 10.9 ± 1.2 | 0.52 |
| $CD4^+$ | 24.2 ± 1.1 | 24.6 ± 0.5 | 0.77 | 22.6 ± 2.5 | 21.7 ± 1.4 | 0.77 |
| $CD4^+CD62L^+CD44^+$ | 8.2 ± 0.7 | 6.4 ± 0.8 | 0.13 | 8.4 ± 0.6 | 7.6 ± 0.4 | 0.30 |
| $CD8^+$ | 14.3 ± 1.3 | 12.6 ± 0.3 | 0.24 | 13.3 ± 1.7 | 11.4 ± 0.9 | 0.38 |
| $CD8^+CD62L^+CD44^+$ | 16.9 ± 1.2 | 9.7 ± 0.7 | 0.13 | 18.7 ± 1.2 | 11.6 ± 0.1 | 0.003 |
| $B220^+$ | 37 ± 5.5 | 46.3 ± 1.2 | 0.15 | 48.5 ± 1.3 | 47.5 ± 3.8 | 0.79 |
| $CD11b^+$ | 9.5 ± 1.9 | 7.0 ± 0.6 | 0.27 | 8.8 ± 0.4 | 5.5 ± 0.6 | 0.003 |
| $CD11b^+Ly6C^+Ly6G^+$ | 41.3 ± 3.1 | 39 ± 2.5 | 0.59 | 45.8 ± 3.9 | 39.2 ± 2.2 | 0.21 |
| RBC ($10^6$/$mm^3$) | 11.5 ± 0.3 | 11.3 ± 0.1 | 0.72 | 12.2 ± 0.7 | 11.5 ± 0.6 | 0.47 |
| WBC ($10^3$/$mm^3$) | 8.3 ± 0.3 | 12.3 ± 1.9 | 0.076 | 10.7 ± 1.2 | 8.2 ± 1.4 | 0.45 |
| Hematocrit (%) | 59.1 ± 3.2 | 55.3 ± 0.7 | 0.29 | 56.5 ± 1.9 | 58.7 ± 3.3 | 0.62 |
| Lymphocytes (%) | 75.5 ± 2.3 | 77.4 ± 2.0 | 0.56 | 75.6 ± 4.0 | 79.6 ± 3.4 | 0.49 |
| Monocytes (%) | 4.5 ± 0.4 | 4.5 ± 0.3 | 0.92 | 4.8 ± 0.7 | 4.3 ± 0.7 | 0.67 |
| Granulocytes (%) | 20.1 ± 1.9 | 18.1 ± 1.8 | 0.49 | 19.6 ± 3.2 | 16.1 ± 3.0 | 0.46 |

deregulated in $CD3^+$ TILs from tumor-bearing Nr2f6-deficient mice additionally treated with anti-PD-L1. In order to proof the in silico analysis of a deregulated lymphocyte activation and metabolic function, we investigated the extracellular acidification mirroring the pH of sorted $CD3^+$ T cells in vitro and were able to observe a significant higher acidification in the media of Nr2f6-deficient T cells reasoning a higher metabolic rate of lymphocytes (Supplementary Fig. 3C).

Taken together, targeting the NR2F6 pathway is a mechanistically independent option in cancer treatment regimens and identifies a candidate role of the orphan nuclear receptor NR2F6 in governing effector T-cell function relevant for cancer cell rejection.

**Heterozygous $Nr2f6^{+/-}$ mice similarly demonstrate the benefit.** To evaluate the effect of partial versus complete Nr2f6-deficiency on tumor growth suppression, we injected wild-type $Nr2f6^{+/+}$, heterozygous $Nr2f6^{+/-}$ and knockout $Nr2f6^{-/-}$ mice with either B16-OVA or MC38 tumor cells and monitored tumor growth. Comparable to the $Nr2f6^{-/-}$ mice, tumor progression was also significantly slower and overall tumor masses smaller in heterozygous $Nr2f6^{+/-}$ mice (Fig. 4a, c). Heterozygous $Nr2f6^{+/-}$ mice showed significant better survival after tumor challenge with both B16 and MC38 cell lines similar to homozygous $Nr2f6^{-/-}$ mice (Fig. 4b, d). Similarly, the treatment of MC38 tumor-bearing heterozygous $Nr2f6^{+/-}$ mice with αPD-L1 is as effective as in knockout mice (Fig. 4e). Infiltrate analyses of tumor-bearing heterozygote animals also demonstrate enhanced numbers of $CD45^+$ cells. Haplo-insufficiency of the Nr2f6 gene function in vivo is shown by the fact that one deficient allele of the Nr2f6 gene was sufficient to increase the immune system's efficacy to counteract tumor outgrowth. Investigation of cytokine and proliferation responses of isolated $CD4^+$ (Fig. 4f) and $CD8^+$ (Fig. 4g) T cells in vitro, albeit only in part, confirmed a functional effect of haplo-insufficiency of the Nr2f6 gene.

**Acute Nr2f6 inhibition is sufficient for hyper-responsiveness.** As previously reported, both murine $CD3^+$ effector T cells (but importantly not regulatory T cells[23]), activated in the absence of NR2F6, exert enhanced effector functions. To confirm the importance of NR2F6 as T-cell-intrinsic suppressor of T-cell-mediated tumor growth control in vivo, we next employed ex vivo siRNA-mediated Nr2f6 silencing prior ACT of autologous T cells

into a MC38 subcutaneous mouse tumor model. Fully immunocompetent wild-type mice were injected with Nr2f6 siRNA or siRNA control transfected polyclonal $CD3^+$ T cells, in combination with PD-1/PD-L1 axis blockade, respectively. Adoptive transfer of $CD3^{Nr2f6\ siRNA}$ polyclonal T cells that demonstrated significant Nr2f6 silencing (Fig. 5a) was sufficient for a significant delay in tumor growth when compared to mice receiving $CD3^{control\ siRNA}$ cells (Fig. 5b–d). Analysis of congenic marked siRNA treated $CD3^{control\ siRNA}$ and $CD3^{Nr2f6\ siRNA}$ T cells in a competitive adoptive transfer experiment revealed significantly elevated IL-2 levels in Nr2f6 siRNA transfected $CD4^+$ T cells in the draining lymph node of mice receiving PD-L1 blockade therapy. In keeping with this finding, siRNA-mediated human NR2F6 silencing induced hyper-responsiveness, as measured by significantly augmented IFNγ and IL-2 responses, thus validating an essential and non-redundant function of NR2F6 in human T cells (Fig. 5e-j), a result highly reminiscent of the murine Nr2f6 knockout T cells ex vivo. Adoptively transferred Nr2f6 silenced $CD3^+$ T cells therefore act as robust "sensitizer" for the established anti-PD-L1 immune checkpoint blockade in mouse tumor models in vivo.

Of note, although the NR2F family consists of three orphan receptors (designated as NR2F1, NR2F2 and NR2F6), only NR2F2 and NR2F6 were found to be expressed in primary human T cells (Supplementary Fig. 5A, B). Within the NR2F family, NR2F6 has an isotype-selective role in immune cells as is shown by strongly enhanced expression of NR2F6 (but not NR2F2) upon CD3/CD28-induced human T-cell stimulation. Human NR2F6 mRNA expression levels increased after CD3/CD28 stimulation, reaching levels of up to a fivefold (for $CD4^+$ T cells) or up to a tenfold (for $CD8^+$ T cells) augmentation (Supplementary Fig. 5A,B). Additionally, no compensatory upregulation of NR2F2 mRNA as a consequence of NR2F6 siRNA-mediated silencing was observed (Supplementary Fig. 5C, D). Finally, in contrast to NR2F6, siRNA-mediated human NR2F2 silencing did not induce any augmented IFNγ and IL-2 responses (Supplementary Fig. 5E). Ex vivo siRNA-mediated Nr2f6 silencing and stimulation of murine $CD8^+$ T cells showed elevated IFNγ and IL-2 responses measured by qRT-PCR (Supplementary Fig. 5F) and FACS analysis (Supplementary Fig. 5G) detectable up to 7 days after transfection.

Taken together, siRNA-mediated Nr2f6 silencing as a single negative regulator of T-cell signaling augments immune activity

in experimental mouse in vivo and human ex vivo model systems. This defines a non-redundant role of NR2F6 as a negative feedback loop during CD3+ effector T-cell activation.

**Upregulation of NR2F6 in TILs from NSCLC tumor biopsies.** To establish a direct causal relationship between lymphatic NR2F6 and human cancer, intratumoral NR2F6 expression

was determined by immunohistochemical (IHC) analyses of tissue microarrays (TMA). Importantly, NR2F6 protein expression was seen in TILs. NR2F6 expression in TILs was upregulated in more than 50% (164 of 303: 54%) of NSCLC patients (Fig. 6a). When using a staining score from 0 to 4, lung cancer tissue clearly contained higher numbers of NR2F6-expressing TILs when compared to healthy lung tissue (Fig. 6b). Significant higher *NR2F6* mRNA expression could also be seen in sorted CD3+

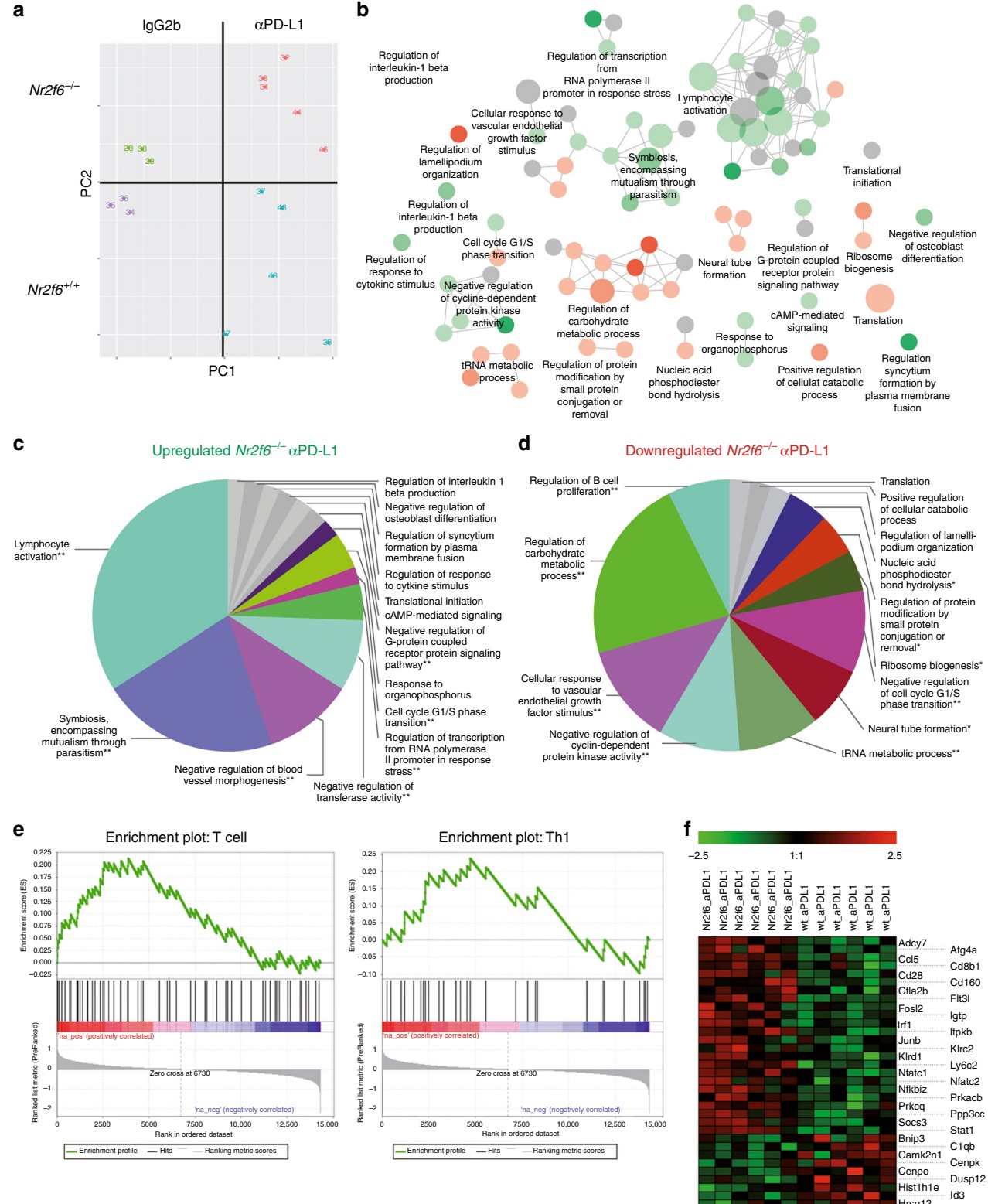

T cells of tumor tissue from NSCLC patients compared to PBMCs (Fig. 6c). However, expression levels of NR2F6 in TILs were not linked to overall survival in the examined patients. This also includes the high (>3–4) NR2F6 expression group (Fig. 6d). Notably, however, the number of NR2F6-expressing TILs was highly correlated to the abundance of PD-1 and CTLA-4-expressing TIL as well as PD-L1 expression on tumor cells (Fig. 6e, Supplementary Fig. 6). These data suggest a potential rationale for combined targeting of NR2F6 and PD-L1/PD-1 in human cancer and may hint at a specific immunological finger-print helping to select or stratify patients for the most appropriate immune-activating therapeutic strategy.

Taken together, NR2F6 expression in TILs of NSCLC may be linked to T-cell dysfunction/exhaustion phenotype that supports its potential suitability as an additive or even synergistic target for immune-activating strategies to be combined with established checkpoint blockade.

## Discussion

The challenge in the field of cancer immunology is to discover the most suitable targets as well as their best combinations for therapeutic intervention. In this study we provide evidence for an essential and non-redundant role of NR2F6 in critically antagonizing anti-cancer effector T-cell responses. We made the remarkable observation that loss of NR2F6 enhances the activity of established PD-L1 checkpoint blockade to promote tumor regression and increased survival in subcutaneous tumor models. The synergistic therapy outcome indicates that NR2F6 blocking is a mechanistically independent option and indicates that inhibitory targeting of NR2F6 may provide novel immunotherapeutic avenues to enhance T-cell checkpoint immunotherapy. Importantly, despite the dramatically improved clinical outcome in $Nr2f6^{-/-}$ tumor-bearing mice, no exacerbated signs of irAE were observed in the combinatorial NR2F6/PD-L1 targeting group when directly compared to the mono-therapy mice, suggesting that major side effects might not hamper the therapeutic potential of this combinatorial approach.

Nuclear receptors are transcription factors that have been shown to be essential for both pro- and anti-inflammatory T-cell responses[24,25]. Of note, nuclear receptors generally regulate gene expression in response to the binding of lipophilic ligands and thus, the nuclear receptor family has a strong history of successful drug discovery[26–28]. There is as yet no valid information about endogenous ligands for several human nuclear receptors including NR2F6[29,30]. However, although the NR2F family members are classified as orphan receptors, their ligand-binding domain

(LBD) is both evolutionarily conserved and functionally indispensable for its transcriptional activity[22,31]. Mutations within the LBD of NR2F6, reducing the size of the ligand-binding pocket or disrupting co-repressor interactions, have shown to significantly reduce its transcriptional repressor function[22], indicating that endogenous ligands for NR2F6 may exist and presumably modulate NR2F6 function.

Along this line of argumentation, intriguingly, even heterozygous $Nr2f6^{+/-}$ mice exhibited the same ability to delay tumor outgrowth of implanted B16-OVA and MC38 tumors as $Nr2f6$ knockout mice. Because complete $Nr2f6$ deficiency, as in homozygous knockout mouse models, hardly represents the physiological level of inhibition that can be reached during clinical regimens, this observed haplo-insufficiency of the $Nr2f6$ gene function in vivo further supports the suitability of NR2F6 as a target for cancer immunotherapy. Thus, not only complete but also partial (50%) NR2F6 deficiency, the latter representing a fairly realistic scenario of therapeutic inhibition, is sufficient to strengthen anti-tumor immunity. In agreement with these observations, acute $Nr2f6$ gene silencing in both mouse and human T cells induced hyper-responsiveness, firmly validating a non-redundant T-cell inhibitory function of NR2F6.

Because T lymphocyte-intrinsic NR2F6 acts as a potent and selective repressor of effective cancer immunity, NR2F6 appears to represent a crucial regulator of cancer immune tolerance induction and maintenance. Mechanistically, as examined by RNA sequencing of TILs derived from tumor-bearing mice, NR2F6 appears to set the threshold of T-cell effector functions as transcriptional regulator of critical target genes shaping activation, recruitment, proliferation and homeostasis of tumor-antigen-specific T-cell responses.

It has been shown that T cells in the TME frequently express high levels of inhibitory receptors as a sign of T-cell exhaustion, including PD-1, CTLA-4, as well as impaired effector cytokine production, such as IL-2, TNFα, IFNγ. Interestingly, this impairment of cytokine production would correlate with high levels of NR2F6 expression, as IL-2, TNFα and IFNγ are direct target genes of NR2F6-dependent transcriptional repression[23]. Along this line of argumentation, a positive correlation between NR2F6 expression and cancer has been reported; studies have identified NR2F6 to be upregulated in human ovarian cancer, colon cancer and lymphoma[32–38]. However, it is not possible from these published data to determine whether the origin of NR2F6 expression is the malignant or the immune cell. Notably, in the tumor-infiltrating T cells of 54% of lung cancer patients, NR2F6 upregulation has been observed within the TME. This suggests that targeting the immune checkpoint NR2F6 in those

**Fig. 3** Nr2f6 expression alters gene signature of tumor-reactive T cells. **a** Principal component analyses of the RNA-seq data from pre-sorted CD3+ tumor-infiltrating T cells of $Nr2f6^{+/+}$ IgG2b ($n = 3$), $Nr2f6^{+/+}$ αPD-L1 ($n = 5$), $Nr2f6^{-/-}$ IgG2b ($n = 3$) and $Nr2f6^{-/-}$ αPD-L1- ($n = 5$) treated tumor-bearing mice taken at d14 after tumor injection of $5 \times 10^5$ B16-OVA melanoma cells separates the TILs into four distinct clusters. **b**–**d** ClueGO analysis of up- and downregulated genes in isolated TILs from combinatorial NR2F6/PD-L1 therapy groups. CD3+ TILs from either $Nr2f6^{+/+}$ or $Nr2f6^{-/-}$ mice with PD-L1 blockade therapy were isolated, RNA-seq was performed and the significantly differentially expressed genes were subsequently analyzed using ClueGO. The enriched gene ontology terms are shown as functionally grouped nodes in an interconnected network based on their score level. The sizes of the nodes reflect the enrichment significance of the terms, while functionally related groups partially overlap. Terms with up-/downregulated genes are shown in green/red, respectively. The color gradient shows the gene proportion of each group (up- or downregulated group of genes) associated with the term. Equal proportions of the two groups are represented in gray. The pie charts show the enriched groups represented by the most significant term. The sizes of the sections correlate with the number of terms included in a group. The key upregulated pathways (**c**) in TILs from $Nr2f6^{-/-}$ mice with PD-L1 blockade are: cell-mediated cytotoxicity ($p = 0.0001$), IFNγ signaling ($p = 0.0004$), TCR signaling pathway ($p = 0.0006$), immune system function ($p = 0.0009$), Wnt signaling pathway ($p = 0.0029$), type II IFN signaling ($p = 0.0042$), TNF signaling pathway ($p = 0.0045$). **e** GSEA enrichment plot of KEGG endometrial cancer pathway genes of sorted CD3+ TILs from $Nr2f6^{+/+}$ αPD-L1 versus $Nr2f6^{-/-}$ αPD-L1-treated tumor-bearing mice. Genes in the KEGG endometrial cancer signaling pathway showed significant enrichment in T cells and CD4 Th1 signature (FDR $q$ value=0.006). The top portion of the figure plots the enrichment scores (ES) for each gene, whereas the bottom portion of the plot shows the value of the ranking metric moving down the list of ranked genes. **f** Heat map showing most prominent deregulated genes: Nfatc1; Nfatc2; Klrc2; Cd48; Klra3; Ppp3cc; Klrd1; Socs3; Stat1; Camk2b; Cd28; Il10; Prkcq; Adcy7; Hectd3; H2-DMa; Dapp1; Rps6kb2; Cybb; Ripk3; Tnfaip3; Prkacb; Zbp1; Junb; Ccl5

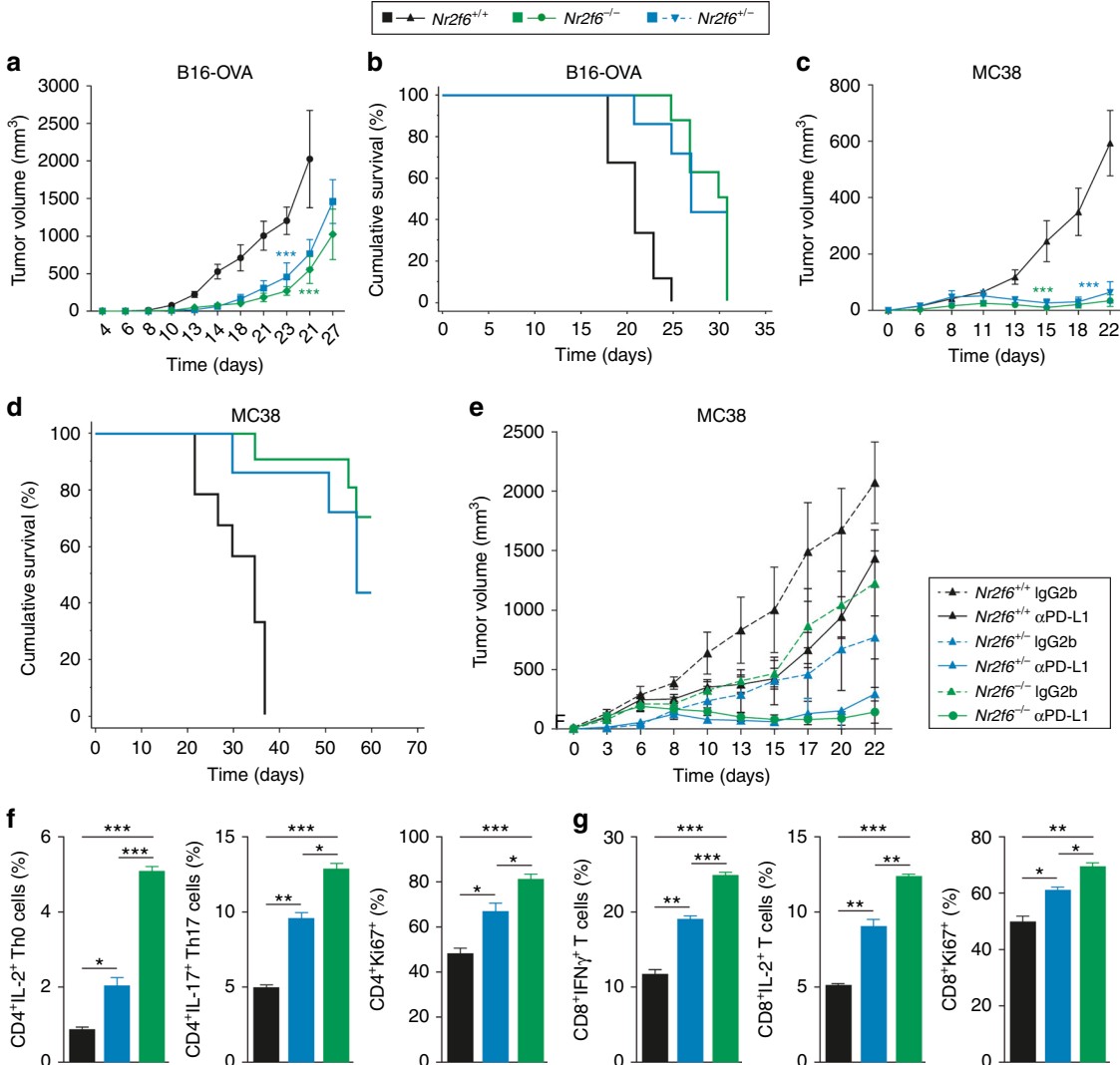

**Fig. 4** Heterozygous $Nr2f6^{+/-}$ mice similarly show strong tumor growth benefit. Tumor growth curves of $Nr2f6^{+/+}$, $Nr2f6^{+/-}$ and $Nr2f6^{-/-}$ mice that received **a** 1×10[5] B16-OVA melanoma cells ($Nr2f6^{+/+}$ ($n = 21$), $Nr2f6^{+/-}$ ($n = 11$), and $Nr2f6^{-/-}$ ($n = 21$)) or **c** 5×10[4] MC38 tumor cells ($Nr2f6^{+/+}$ ($n = 16$), $Nr2f6^{+/-}$ ($n = 5$), and $Nr2f6^{-/-}$ ($n = 10$)) subcutaneously and were monitored three times a week showing a significant tumor growth delay as well as a survival benefit (**b-d**) in $Nr2f6$ gene-modulated mice ($p<0.0001$ for both tumor cell lines, ANOVA). **e** Tumor growth curves of $Nr2f6^{+/+}$ (IgG2b $n = 5$, αPD-L1 $n = 10$), $Nr2f6^{-/-}$ (IgG2b $n = 7$, αPD-L1 $n = 12$) and $Nr2f6^{+/-}$ (IgG2b $n = 3$, αPD-L1 $n = 3$) mice injected s.c. with 5×10[5] B16-OVA melanoma cells and treated with PD-L1 blockade (continuous line) or IgG2b (dashed line) ($n = 3-5$). Both αPD-L1 treated $Nr2f6^{+/-}$ ($p = 0.0179$, ANOVA) and $Nr2f6^{-/-}$ ($p < 0.0001$, ANOVA) showed a significant slower tumor growth when compared to treated wild-type controls. **f**, **g** In vitro flow cytometry analysis of isolated CD4[+] or CD8[+] T cells activated with anti-CD3 mAb (5 μg) and anti-CD28 mAb (1 μg) at d3 from $Nr2f6^{+/+}$, $Nr2f6^{+/-}$ or $Nr2f6^{-/-}$ mice ($n = 3$). Analysis of IL-2 and IL-17-producing CD4[+] Th0 and Th17 T cells, Ki67[+] proliferating CD4[+] and CD8[+] T cells, and IFNγ as well as IL-2 production of CD8[+] T cells. Percentage of positive cells relative to parental gate is shown; experiments were repeated at least three times. Results shown are derived from at least two independent experiments. Error bars represent the mean ± SEM

presumably tumor antigen-reactive but probably exhausted T-cell clones in the patient group with NR2F6[Positive] TILs may result in increased anti-tumor effector responses during an envisioned NR2F6-targeted therapy.

According to all our preclinical data, high NR2F6 expression is likely to critically contribute to the immune-suppressed state of tumor antigen-specific TILs, thus limiting the host's anti-tumor immune response. Absence of NR2F6, either through genetic ablation in $Nr2f6$ knockout mice or by the means of siRNA knockdowns, increases the immune system's ability to mount effective immune responses, resulting in striking anti-tumor effects in different advanced mouse models relevant to human cancer.

This preclinical proof of concept study on the $Nr2f6$ gene depletion-mediated immune augmentation is envisioned as a way

forward to extend the benefits of clinical immuno-oncology therapies to a larger number of cancer patients. The unique feature of lymphatic NR2F6 as an alternative immune checkpoint may influence cancer therapy in the future, both in terms of targeting cancer pathways using small molecule inhibitors, and individualized adoptive therapy of $NR2F6$ gene-modified human T cells, a hypothesis that now requires further validation.

## Methods

**Mice.** $Nr2f6$-deficient mice[39] back-crossed eight times on C57BL/6 background were used. Mice were maintained under SPF conditions. All animal experiments were performed in accordance with national and European guidelines and reviewed and authorized by the committee on animal experiments (BMWFW-66.011/0064-WF/V/3b/2016). Investigations were strictly gender-stratified and not blinded.

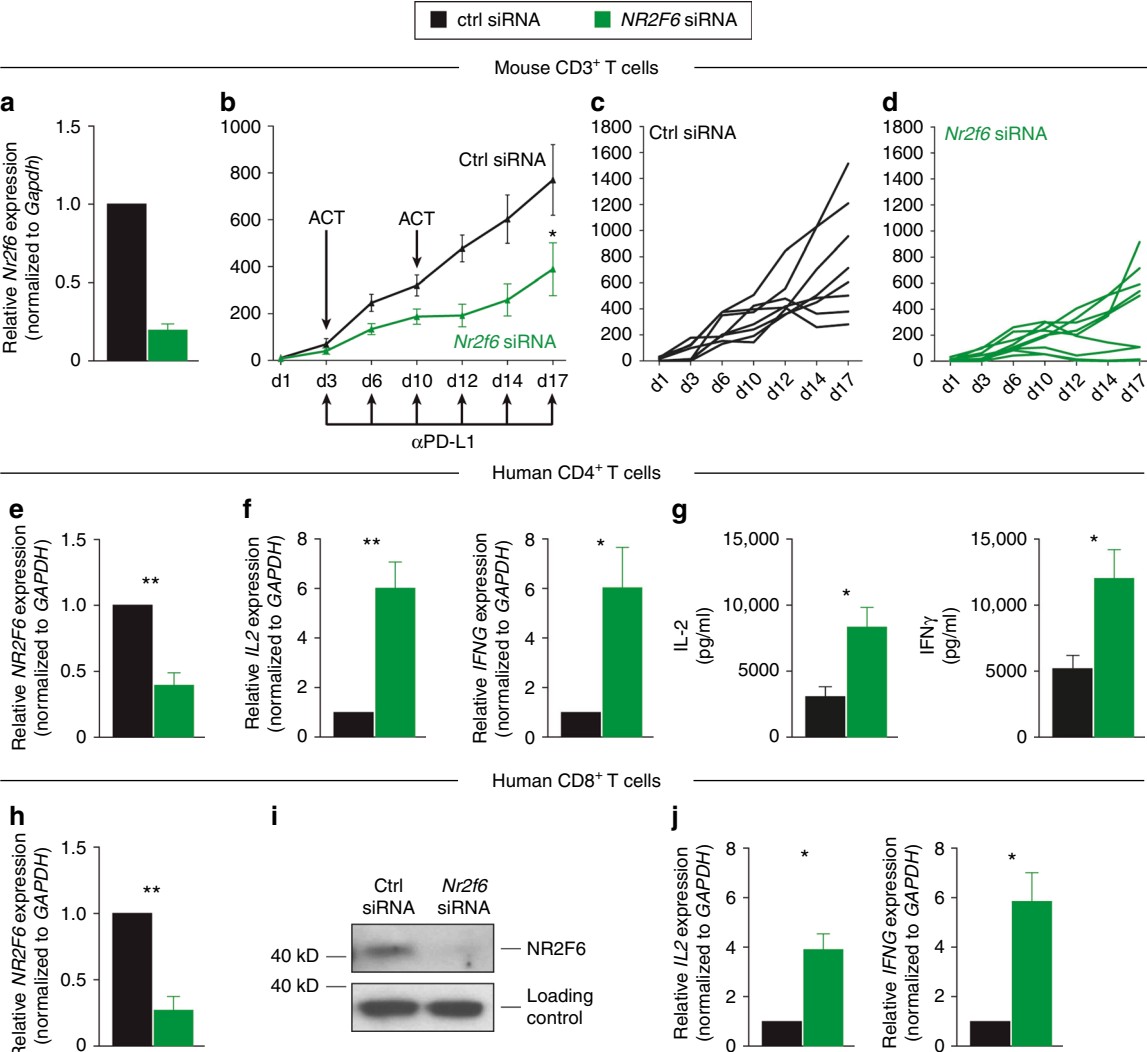

**Fig. 5** *Nr2f6* gene silencing is effective and leads to hyper-responsiveness of T cells. **a−d** Tumor growth effects depend primarily on NR2F6 function in CD3[+] T cells. **a** Mouse *Nr2f6* gene silencing is effective in CD3[+] T cells ($p = 0.002$, unpaired *t*-test). Mouse CD3[+] T cells were nucleofected with *Nr2f6* siRNA or *control* siRNA as indicated. Silencing efficacy of *Nr2f6* siRNA was analyzed by qRT-PCR and normalized to *Gapdh* ($n = 3$). **b−d** Wild-type recipients received 5×10[5] MC38 tumor cells s.c. and were treated with therapeutic adoptive cell transfers (ACT) of 1×10[7] CD3[+] T cells transfected with *control* (black, $n = 8$) or *Nr2f6* (green, $n = 8$) siRNA on d3 and d10 after tumor injection ($p = 0.0172$, ANOVA). Mice were also treated with 0.25 mg of antibody against PD-L1 (Ab10F.9G2) on the respective days. Single tumor growth curves of wild-type mice receiving ACT of **c** *control* siRNA transfected CD3[+] T cells or **d** *Nr2f6* siRNA silenced CD3[+] T cells. In the *Nr2f6* siRNA ACT group, one mouse (11.1%) was able to reject the high tumor burden completely, but all mice of the control siRNA ACT group had to be killed latest on d19. **e−j** Human CD4[+] and CD8[+] T cells were nucleofected with *NR2F6* siRNA or control siRNA as indicated ($n = 3$). Silencing efficacy of *Nr2f6* siRNA was analyzed by qRT-PCR (**e** CD4[+] $p = 0.0037$, **h** CD8[+] $p = 0.002$) and immunoblotting (**i**) of CD8[+] T cells. Cytokine production was measured by qRT-PCR after 5 h of PDBu (0.05 μg/ml)/ionomycin (0.5 μg/ml) stimulation and showed significantly elevated levels of *IL2* ($p = 0.009$) and *IFNG* ($p = 0.04$) in CD4[+] T cells (**f**) as well as *IL2* ($p = 0.046$) and *IFNG* ($p = 0.05$) in CD8[+] T cells (**i**). **g** Bioplex protein analysis also showed elevated levels of IFNγ ($p = 0.019$) and IL-2 ($p = 0.017$) in human CD4[+] T cells transfected with *Nr2f6* siRNA. Similar results were obtained with non-overlapping *Nr2f6* siRNA oligonucleotides. Results shown are derived from at least two independent experiments. Data are presented as the mean ± SEM, analyzed by a two-tailed unpaired Student's *t*-test

Experimental mice were randomly chosen from litters with a minimal sample size of three.

**Tumor induction and adoptive cell transfer**. 1×10[5] B16-OVA, 5×10[4] MC38 (kindly provided by Maximillian Waldner, University of Erlangen, Germany) were injected subcutaneously (s.c.) into the left flank of 8- to 12-week-old wild-type, *Nr2f6*[+/−] or *Nr2f6*[−/−] mice. For in vivo antibody blockade experiments, a higher tumor load was used by s.c. injection of 5×10[5] B16-OVA or 7.5×10[5] MC38 tumor cells. The high tumor load (5×10[5]) was applied with therapy (anti-PD-L1 or ACTs) to ensure robust tumor growth. Tumor growth was monitored three times a week by measuring tumor length and width. Tumor volume was calculated according to the following equation: ½(length × width²). For survival analysis, mice with tumors greater than the length limit of 15 mm were sacrificed and counted as dead. Cell lines were tested negative for mycoplasma (GATC, Konstanz, Germany).

**In vivo antibody blockade**. Mice were injected s.c. with 5×10[5] B16-OVA melanoma cells or 7.5×10[5] MC38 tumor cells and administered either 0.5 mg (B16-OVA) or 0.25 mg (MC38) of anti-mouse PD-L1 (Clone10F.9G2; BE0101) or corresponding IgG2b (LTF-2; BE0090), control (all from BioXCell, USA) every 3 days starting from day 1 or day 3 of tumor challenge according to ref.[40]. For CD4[+] and CD8[+] depletion experiments, mice were administered 0.05 mg of anti-mouse CD4 (Clone GK1.5), anti-mouse CD8 (clone YTS169.4) or corresponding IgG2b (LTF-2; BE0090) 5 days and 3 days before s.c. injection of 1×10[5] B16-OVA melanoma cells as well as 5 and 12 days after tumor challenge.

**MCA carcinogenesis model**. Groups of female wild-type and *Nr2f6*[−/−] mice were inoculated s.c. in the hind leg with 200 mg of 3-MCA (Sigma-Aldrich) in 0.1 ml of corn oil as described[41]. Mice were then monitored weekly for fibrosarcoma development over 300 days and recorded as percentage of tumor-free mice.

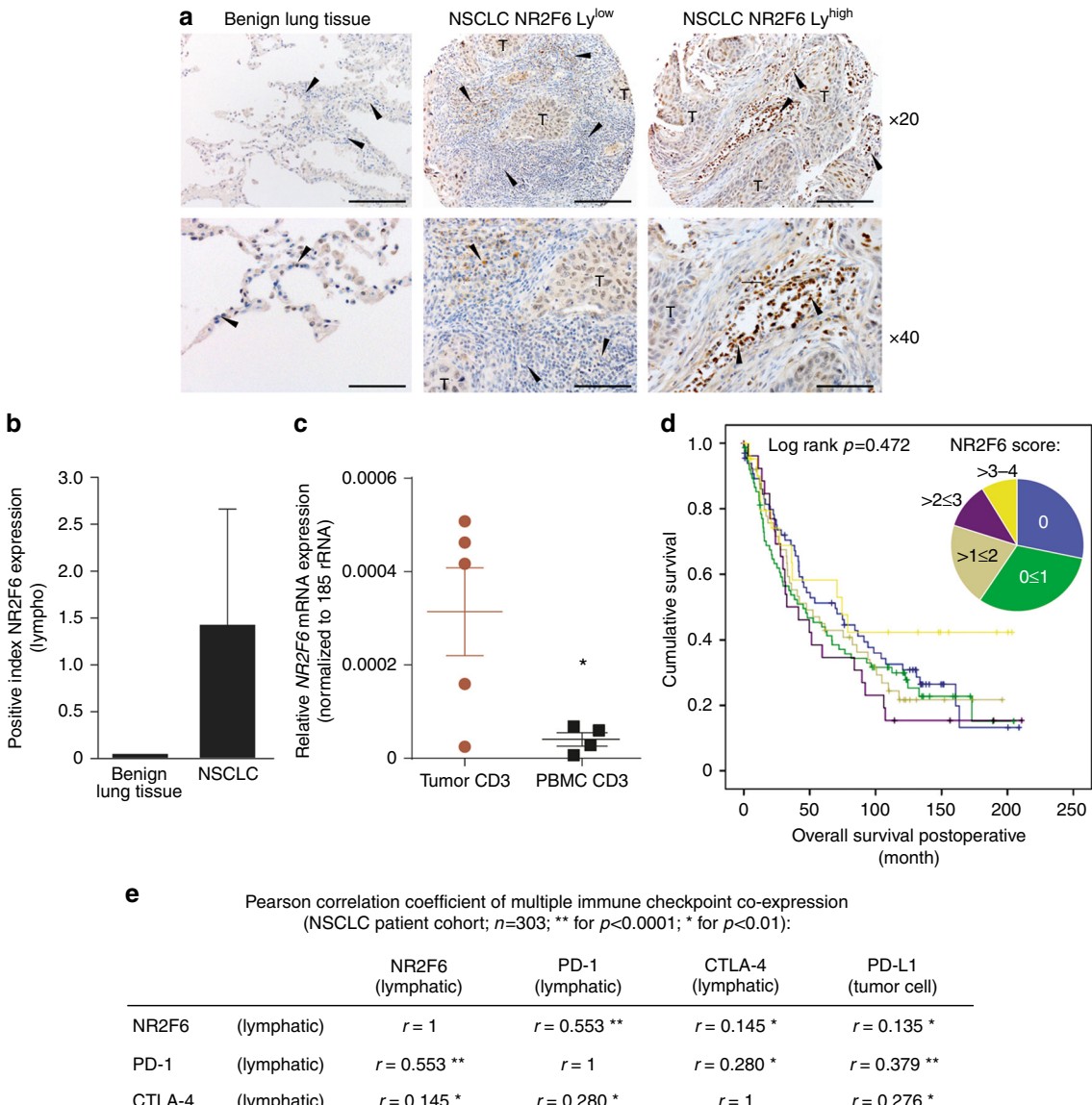

**Fig. 6** IHC and qRT-PCR analyses of NSCLC tumor biopsies show upregulation of NR2F6 in TILs that may serve as marker of patient stratification. **a** In patient tumor biopsies, there is significantly upregulated expression of NR2F6 in T cells (marked by arrows) in 54% of the cases (163 of 303 patients). **b** There is a significantly enhanced expression of NR2F6 in TILs from NSCLC tumor biopsies when compared to benign lung tissue ($n = 303$ and $n = 10$, $p <$ 0.001). **c** Significant higher *NR2F6* expression levels ($p = 0.038$, unpaired *t*-test) determined by qRT-PCR in $CD3^+$ cells sorted from NSCLC tumor biopsies when compared to PBMC samples. **d** The number of NR2F6-expressing TILs was defined as categorial variable: $0 = 0\%$, $1 = 1-25\%$, $2 = 26-50\%$, $3 = 51-75\%$ and $4 = 76-100\%$ NR2F6-positive TILs. The number of NR2F6-expressing TILs is not linked to clinical outcome as seen in overall survival of NSCLC patients from these defined staining categories (0–4). **e** Multiplex IHC analyses showed that the increase in lymphatic NR2F6 expression strongly correlated with lymphatic PD-1 expression (Pearson correlation coefficient $r = 0.553$, $p < 0.001$). There was a positive correlation also between lymphatic NR2F6 and lymphatic CTLA-4 ($r = 0.145$, $p < 0.01$) and tumor cell PD-L1 ($r = 0.135$, $p < 0.01$) expression, respectively

Palpable tumors 10 mm² with progressive growth for two successive weeks were recorded.

**3-MCA tumor-induced cell lines**. Sarcomas that had reached a size of 0.5 cm² were excised aseptically. Tumors were cut into small pieces and treated with 1 mg/ml collagenase (type II; Sigma-Aldrich) and 40 µg/ml DNase I at 37 °C for 1 h, clumps were removed, and single cells were cultured in RPMI 1640 complete medium (10% FCS, 2 mM L-glutamine, and 50 U/ml penicillin/streptomycin). The cells were split with Trypsin/EDTA (T3924 Sigma) when they were confluent. All tumor cell lines were kept in culture for at least 2 months to minimize cellular contamination. Some sarcoma lines were injected s.c. into wild-type or *Nr2f6*-deficient mice as indicated.

**Preparation of tumor-infiltrating cells**. Mononuclear infiltrating cells were isolated from both subcutaneous and chemically induced tumors at the indicated time

points. Briefly, tumor tissues from sacrificed mice were prepared by mechanical disruption followed by digestion for 45 min with collagenase D (2.5 mg/ml; Roche, 11088858001) and DNase I (1 mg/ml; Roche, 11284932001) at 37 °C. Digested tissues were incubated 5 min at 37 °C with EDTA (0.5 M) to prevent DC/T-cell aggregates and mashed through a 100-µm filter and a 40-µm filter. Cells were washed, and resuspended in either PBS+ 2% FCS or RPMI complete medium (10% FCS, 2 mM L-glutamine, and 50 U/ml penicillin/streptomycin) and used for subsequent experiments.

**Flow cytometry**. Splenocytes were depleted of erythrocytes using the mouse erythrocyte lysing kit (R&D, WL2000) and, like lymph node cells, mashed through a 100-µm filter. Splenocytes, lymph node cells and TILs were incubated with FcR Block (BD Biosciences, 553142) to prevent nonspecific antibody binding before staining with appropriate surface antibodies for 30 min at 4 °C, washed with PBS+

2% FCS, and used for FACS analysis. For intracellular cytokine staining, cells were stimulated with 50 ng/ml phorbol 12,13-dibutyrate (PDBu, Sigma, P1269), 500 ng ionomycin (Sigma, I0634) and GolgiPlug (BD Biosciences, 555029) for 4–5 h. After fixation (cytokines: Biolegend fixation buffer (420801), 20 min, 4 °C; transcription factors: FoxP3 staining buffer set (eBiosciences, 00-5523), >30 min, 4 °C), cells were permeabilized with the fixation/permeabilization kit (BioLegend, 421002) for cytokines and the Foxp3-staining buffer set (eBiosciences, 00-5523) for transcription factors, incubated with FcR Block (BD Biosciences, 553142) before staining with specific cell surface or intracellular marker antibodies. Data were acquired on an FACSCalibur, LSR Fortessa or FACS Verse cell analyzer (Becton Dickinson). Data were analyzed by FlowJo software (version 10). The following antibodies were used for flow cytometry: CD4-V500 (BD, 560783), CD4-PE (BD, 553049), CD8a-APC (BD, 553035), CD25-PE (BD, 553866), CD44-FITC (BD, 553270), CD62L-APC (BD, 553152), PD-1-PE (BD, 561788), Ki67-PE (BD, 556027), IL-17-PE (BD, 559502), IL-2-APC (BD, 554429), CD45-V500 (BD, 561487), CD45-FITC (eBiosciences, 11-0451-82), CD8a-PE (eBiosciences, 12-0081-82), CD8a-PerCP Cy5.5 (eBiosciences, 45-0081-82), CD44-PerCP Cy5.5 (eBiosciences, 45-0441-82), IFNγ-PE-Cy7 (eBiosciences, 25-7311-82), PDL-1-PerCP-eFluor710 (eBiosciences, 46-5982-80), CD45-APC (eBiosciences, 17-0451-81), CD3-PE-Cy7 (eBiosciences, 25-0031,82), FoxP3-FITC (eBiosciences, 11-5773-82), CD3-PE (eBiosciences, 12-0031-83), CD8a-bv421 (BioLegend, 100738), CD25-bv421 (BioLegend, 102034).

**siRNA Nr2f6 inhibition in vitro**. Isolated mouse or human T cells as described above were used for transfection by the established method, namely siRNAs employing Lonza® Nucleofection Technology using the human T-cell nucleofector kit from Amaxa (Lonza, VPA-1002). Transfected mouse T cells were incubated in complete RPMI medium supplemented with 2.5 ng/ml IL-7 (RnD, 407-ML), 2 ng/ml IL-2 (eBiosciences, 14-8021-64) and 50 ng/ml IL-15 (Biolegend, 566302) for 48 h before further restimulation. The used siRNA sequences for mouse: siRNA J-054088-11, Nr2f6, GCAUCGACAACGUGUGCGA, and human: siRNA J-003423-06 NR2F6, CGGCAAGCAUUACGGUGUC (both from ThermoScientific).

**siRNA Nr2f6 inhibition and adoptive cell transfer**. $5 \times 10^5$ MC38 tumor cells were injected s.c. into C57BL/6 wild-type recipients. Two adoptive cell transfers (ACT) of control or Nr2f6 siRNA transfected CD3$^+$ T cells into wild-type mice was performed 3 and 10 days after tumor induction by injecting $1 \times 10^7$ sorted CD3$^+$ T cells using the Pan T Cell Isolation Kit II mouse (Miltenyi Biotech 130-095-130) via intra-peritoneal injection. Ab treatment with 0.25 mg anti-mouse PD-L1 (Clone10F.9G2; BE0101) was administered i.p. on d3, d6, d10, d14, d17 and d21.Tumor growth was subsequently measured as described above.

**Immunohistochemistry**. Protein expression in paraffin-embedded NSCLC tissue was assessed on TMA from a cohort provided by the University Hospital Basel containing 405 samples of NSCLC patients from which 303 samples were evaluable[42,43]. For TMA construction, formalin-fixed paraffin-embedded tissues were cut in 4-µm-thick sections and mounted on slides. After staining with hematoxylin and eosin, relevant areas of NSCLC were determined and circled by a pathologist. Four representative cores of the circled regions measuring 0.6 mm in diameter from each formalin-fixed paraffin-embedded NSCLC tissue (donor blocks) were assembled into tissue microarray blocks (recipient blocks) using a semiautomatic tissue arrayer (Beecher Instruments, Sun Prairie, WI, USA). Immunohistochemical (IHC) staining was performed using the Ventana Discovery automated staining system (Ventana Medical System, Tuscon, AZ, USA). Slides were incubated at room temperature with the following primary antibodies: anti-NR2F6 rabbit polyclonal (1:100, ab137496, abcam, Cambridge, UK), anti-PD1 mouse monoclonal (1:100, ab52587, abcam, Cambridge, UK), anti-PD-L1 [28-8] rabbit monoclonal (1:100, ab205921, abcam, Cambridge, UK), anti-CTLA-4 mouse monoclonal (1:100, sc376016, Santa Cruz Biotechnology, Santa Cruz, CA), anti-CD4 mouse monoclonal (1:50, 4B12, Leica Biosystems, Wetzlar, Germany) and anti-CD8 mouse monoclonal (1:40, SP57, Leica Biosystems, Wetzlar, Germany). Detection of the primary antibody was done with the ultraView Universal DAB detection kit (Ventana Medical System, Tuscon, AZ, USA). NR2F6-, PD1, CTLA-4, CD4- and CD8 expression in lymphocytes was assessed by two independent pathologists. NR2F6 and CTLA-4 in lymphocytes were categorized in positive or negative staining, and scored by numbers of positively stained TMA cores per patient (score 0–4). PD-1, CD4, and CD8 were scored by numbers of positive-stained lymphocytes per core (score 0–4 with 1 = 1–25%, 2 = 26–50%, 3 = 51–75%, 4 = 76–100%). PD-L1 staining intensity in tumor cells ranged from 0 to 3 (0 = no staining, 1 = weak staining, 2 = moderate staining, 3 = high staining), positive-stained cells were calculated as 0 = no positive cells, 1 = 1–25%, 2 = 26–50%, 3 = 51–75%, 4 = 76–100%. PD-L1 was scored by immunoreactive score (staining intensity × positive-stained cells).

**Human whole blood samples**. Human T cells were obtained from human peripheral blood, which we get from the local blood bank. Informed consent from all subjects was obtained by the Central Institute for Blood Transfusion at the University Hospital Innsbruck. Mononuclear cells were isolated from human peripheral blood by density centrifugation using Ficoll Buffer (VWR, 17-5442-02).

Magnetically sorted T cells (CD4: 130-045-101; CD8: 130-096-495, Miltenyi Kits) were used for transfection or stimulation.

**Human NSCLC samples**. Five lung adenocarcinoma patients were analyzed for their expression of NR2F6 on CD3$^+$ TILs from single cell suspensions of primary tumors and peripheral mononuclear cells. Approval by the regional ethical board and written IC were given for all patients (AN 2014-0293342/4.5). Four out of five patients were female, median age at diagnosis was 64 years (min 58; max 79). One patient was diagnosed in pUICC stage IB, one in IIA, two IIB and one in stage IIIA. All underwent radical surgical procedures in curative intent without prior neoadjuvant therapies. Thawed cells from tumors and PBMC were stained with antibodies against the following antigens: EpCam (FITC, Miltenyi 130-080-301); CD45 (V450, BD Biosciences 560367); CD3 (BV480, BD Biosciences 566166), and EpCam negative, CD45 and CD3 positive cells were sorted directly into RLT buffer on a FACSAria I (BD Biosciences).

**RNAseq at the i-med DeepSeq Core Facility**. $5 \times 10^5$ B16-OVA tumor cells were injected s.c. into wild-type or Nr2f6-deficient mice. Ab treatment with 0.5 mg anti-mouse PD-L1 (Clone10F.9G2; BE0101) or control Ab IgG2b (LTF-2; BE0090) was injected i.p. starting on d3 twice a week. On d14, tumors were harvested and digested as described above. T cells of Nr2f6 wild-type and Nr2f6$^{-/-}$ mice were pre-sorted with positive beads for CD4 (130-049-201) and CD8 (130-049-401) T cells using kits from Miltenyi and pooled. Isolated T cells were stained with fluorochrome-labeled antibodies recognizing mouse CD3 (17-0031-82, eBioscience). To exclude dead cells, 12.5 µg/ml DAPI was added. Cells were sorted on a BD FacsAria$^{TM}$III Cell sorter (BD Biosciences) and directly collected in RLT buffer (Qiagen) with RNAsin (N2515, Promega). RNA was isolated using the RNeasy® Mini Kit (205113, Qiagen). RNA integrity was verified with the Agilent Bioanalyzer. Ion Torrent$^{TM}$ compatible libraries were generated from 20 ng total RNA input, using the QuantSeq 3′ mRNA-Seq Library Prep Kit from Lexogen (Lexogen, Vienna Biocenter, Austria). Barcoded libraries were multiplexed and sequenced on the Ion Torrent$^{TM}$ Proton Sequencer (Ion torrent, Thermo Fisher Scientific). The Ion Torrent Suite was used for splitting of reads into barcodes, and initial quality filtering and adapter trimming, resulting in a final output of an average 28 million reads per sample. Data analysis has been performed blinded on a collaborative basis with Prof. Trajanoski at the local institute for bioinformatics (http://icbi.i-med.ac.at/). The reads were first preprocessed through a high stringency quality control pipeline consisting of adapter removal with Cutadapt[44] and quality trimming with Trimmomatic[45] to remove bases with bad quality scores. All reads shorter than 22 nucleotides were removed. The quality-trimmed reads were then mapped to the mm10 reference genome using a two-step alignment method; alignment with STAR[46] followed by alignment of the unmapped reads with Bowtie2. From the reads that mapped to multiple locations in the genome, only the primary alignment was retained. Gene-specific read counts were calculated using HTSeq-count. Differential expression analyses were performed using the R package DESeq2[47]. The p-values were adjusted for multiple testing based on false discovery rate (FDR) using the Benjamini−Hochberg approach. Analysis and visualization of Gene Ontology terms associated with differentially expressed genes was performed using ClueGO[48]. Both groups of genes (up and downregulated, p-value <0.05) were used as dual input for GO analysis. The biological terms are grouped together based on their shared genes where the similarity between terms is calculated using kappa statistics. The most significant term was chosen as a representative of the group (Benjamini−Hochberg correction). Gene set enrichment analysis (GSEA)[49] was used to identify immune cell types that are overrepresented in a phenotype. Genes were ranked based on the log2FoldChange and used as an input to pre-ranked GSEA together with a collection of metagenes that correspond to distinct tumor-infiltrating leukocyte subtypes[50].

**Ex vivo T-cell analysis**. CD8$^+$ or CD4$^+$ T cells were isolated using either the mouse CD8 or CD4 T Cell Isolation Kit II or the human CD8 or CD4 T Cell Isolation Kit (Miltenyi Biotec). CD8$^+$ or CD4$^+$ T cells were activated in complete RPMI medium in the presence of plate-bound 2C11 for mouse or Okt3 for human (αCD3, 5 µg/ml) and soluble mouse or human αCD28 (1 µg/ml). Naïve mouse CD4$^+$ T cells were isolated using the CD4$^+$ CD62L$^+$ T cell isolation kit II (Miltenyi). Polarization of these CD4$^+$ T cells into Th17 cells was performed in complete IMDM medium supplemented with TGFβ (5 ng/ml), IL-6 (20 ng/ml), IL-23 (10 ng/ml), anti-IFNγ, and anti-IL-4 (2 µg/ml). Cells were harvested at the indicated time points.

**Gene expression analysis**. Total RNA was isolated using the RNeasy® Mini Kit (Qiagen). First-strand cDNA synthesis was performed using oligo(dT) primers (Promega) with the Qiagen Omniscript RT kit, according to the instructions of the supplier and as described previously[22]. Expression analysis was performed using real-time PCR with an ABI PRIM 7000 or ABI PRIM 7500fast Sequence Detection System with TaqMan gene expression assays (Applied Biosystems); all mouse expression patterns were normalized to Gapdh; all human expression patterns were normalized to GAPDH or EEF1A2.

**irAE method**. $1\times10^4$ MC38 (purchased from the ATCC) were injected subcutaneously (s.c.) into the left flank of 8-week-old female wild-type or $Nr2f6^{-/-}$ mice. In vivo antibody blockade was utilized administering 0.25 mg of anti-mouse PD-L1 (Clone10F.9G2; BE0101) or corresponding IgG2b (LTF-2; BE0090) i.p. twice a week in the first month, and once a week in the following two months. Tumor growth was monitored one to three times a week by measuring tumor length and width. Weight was measured every 2 weeks and small blood samples were taken three times (beginning, middle and end of experiment) for whole blood count analysis using a Vet abc classic hematology system.

**Western blotting**. Cells were washed, and lysed in lysis buffer. Whole-cell extracts or nuclear extracts were electrophoresed on NuPAGE gels (Invitrogen) and transferred to PVDF membranes. Protein lysates were subjected to immunoblotting with antibodies against NR2F6 (FisherScientific Inc., USA: Proteintech 2H2B8, 60117-2, 1:1000), and Actin (Santa Cruz Biotechnology Inc., USA: sc-1615, 1:1000). The uncropped scans are shown in supplementary information (Supplementary Fig. 7).

**Extracellular acidification assay**. $CD3^+$ T cells were stimulated with anti-CD3/anti-CD28 for 3 days in RPMI medium and maintained in a $CO_2$-free incubator at 37 °C for 3 h prior to measurement. Growth medium was removed and cells were washed with respiration buffer. 150 μl of respiration buffer containing pH probe was added. Reagents were preheated to measurement temperature prior to use. All plates were measured kinetically on a multilabel plate reader (PHERAstar, BMG Labtech) at 37 °C for a minimum of 30 min in time-resolved fluorescence (TR-F) mode using a standard Europium filter set ($340 \pm 50$ nm excitation and $615 \pm 8.5$ nm emission). Two TR-F intensity signals were measured at delay times of 100 and 300 μs and a measurement window of 30 μs. Using Microsoft Excel, these readings were converted to lifetime value $s = (t1-t2)/\ln(D1/D2)$, where $s$ is lifetime, $t$ is delay time, and $D$ is intensity values. ECA was assessed as the rate of increase of probe signal (reflecting a decrease in [H+]). OA (Oxamic acid, 1 μM) was used as a negative control. The MitoXpress®—pH Xtra™ Glycolysis Assay Kit from Luxcel (Cat No. PH-100) was used.

**Statistics**. Data were analyzed using Prism 5.03 software (GraphPad Software). Experiments were repeated at least two times with a minimum sample size ($n$) of three. Data are represented as indicated (either the mean ± SEM or ±SD) for all figure panels in which error bars are shown. Overall survival was expressed using the Kaplan−Meier method, and differences between groups were determined using the log-rank test. The $p$-values were assessed using two-tailed unpaired Student's $t$-test, or two-way ANOVA. A $p$ value of <0.05 was considered statistically significant. $*p < 0.05$; $**p < 0.01$; $***p < 0.001$.

**Data availability**. The RNAseq data have been deposited in GEO under the accession number GSE111796 The authors declare that all the other data supporting the findings of this study are available within the article and its Supplementary Information files and from the corresponding authors upon reasonable request.

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

## Acknowledgements

This work was supported by grants from the FFG Austrian Research Promotion Agency (BRIDGE-842388-CBL-AIM) and Austrian Science Fund (MCBO PhD program W1101, P 25440-B21) (G.B.), 30324-B21 (G.B.), P28694-B30 (N.H.-K.) as well as the Austrian Cancer Society, Tyrol (V.K.), the Christian Doppler (CD) Society (G.B.; CD Laboratory I-CARE) as well as the TaNeDS program (DAIICHI SANKYO CO., LTD. Japan). We are grateful to Nina Posch, Nadja Haas and Michaela Kind (all from our institute in Innsbruck) for technical assistance.

## Author contributions

All authors have contributed substantially to the work. V.K. designed the research and performed all mouse in vivo, ex vivo and human ex vivo experiments. N.H.-K., B.J. and P.D.-D. helped with in vivo tumor experiments and analyses of the data. M.E., D.R. and A.K. designed the RNAseq experiments and performed together with Z.T. the bioinformatic analyses. A.T. provided characterized TMA. D.W., A.T. and S.P. outlined and performed the overall experimental design of IHC. A.O. and S.P. performed human IHC experiments. S.S. and G.G. provided human material and performed sorting experiments. All authors critically read the manuscript. G.B. provided the study conception and the overall supervision of the research at all stages.

## Additional information

**Competing interests:** The authors declare no competing interests.

