## [Peer Review File · Nature Communications]

Reviewers' comments:

Reviewer #1 (Remarks to the Author):

In previous works, Baier and Hermmann-Kleiter demonstrated that the nuclear orphan receptor NR2F6 is a PKC substrate and a suppressor of T cell activation and of Th17-dependent autoimmunity using a full germline knockout. Later they published a paper in Cell Reports (2015) demonstrating that a genetic deficiency in NR2F6 delays the growth of cancer cells in a model of prostate cancer and using the B16 melanoma as well. In the last paper, the authors demonstrated by adoptive transfer experiments that NR2F6 deficiency improves the ability of T cells to reject the tumor, indicating that the negative effect of NR2F6 is T cell-intrinsic. In the current paper, Baier and coworkers use a combination of genetic deficiency in NR2F6 with an anti-PDL1 antibody as a demonstration that the combination of an existing checkpoint inhibitor with a future treatment aimed to inhibit NR2F6 would make sense from the point of view of impacting on non-redundant inhibitory pathways. The results are interesting but represent just some incremental value over their previous work in Cell Reports. A piece of information that could make the paper much more relevant could be the expansion of the data contained in Figure 6. In this Figure, the authors try to show that tumor-infiltrating T cells in human lung cancer tumors show upregulated NR2F6 expression indicating that this orphan receptor could be an interesting target for future therapies in humans. However, the data of Figure 6 is supported by immunohistochemistry data which in my opinion is not sufficiently quantitative to sustain the authors' claims. I would recommend that the authors try to generate more data in that direction, isolating TILs from human melanoma or other amenable tumors and using qPCR, intracellular flow cytometry or any other method that can provide real quantitative data.

Reviewer #2 (Remarks to the Author):

The study of Klepsch et al. is built on prior work of the same group characterizing the impact of nuclear orphan receptor NR2F6 deficiency on antitumoral immunity. In prior studies the group had shown that genetic ablation of NR2F6 is associated with an improved tumor control in an autochthonous prostate cancer tumor model and in a syngeneic B16 melanoma model. The current study extends the previous work by investigating NR2F6 ablation in combination with PD-L1 blockade exploiting B16OVA and MC38 syngeneic tumor models. The key finding supported by a set of independent experiments is the synergistic effects of combined NR2F6 ablation PD-L1 blockade.

The studies are well performed, findings are convincing and conclusions sound. The findings are highly relevant and might have a translational impact. The study needs minor revision to be acceptable.

Major comments:

1) The group uses genetically modified mice that were backcrossed to C57/B6. The group should provide information how many rounds of backcrossing were performed. I did not find any information in their previous Cell Report publication, neither.

2) I am intrigued but also surprised that the adoptive transfer of transiently NR2F6 siRNA transfected primary polyclonal CD3 is sufficient to improve tumor control in WT MC28 tumor-bearing mice. I would like to see *in vitro* time kinetic data investigating how long (i) NR2F6 is suppressed in T cells after siRNA knockdown and (ii) this affects functional *in vitro* T cell activation. Since the study is already extensive I do not request additional *in vivo* experiments. However, the authors should discuss potential mechanisms involved in the improved tumor control including priming and effector phase.

Reviewer #3 (Remarks to the Author):

General comment:

The authors demonstrate that loss or reduction of the nuclear factor NR2F6 lowers the threshold for T cell activation in vitro in murine and human T cells. In mice lacking NR2F6, PD-1 blockade was more efficient in different s.c. tumor models. Also adoptive transfer of T cells with NR2F6 silencing reduced tumor growth, a finding that could improve current adoptive cell transfers.

This manuscript reports data of significant value and provides meaningful information on NR2F6 as a candidate target for combinatorial immunotherapies. Yet, the conclusions made by the authors are not completely justified by the results they present. There are several weaknesses and inconsistencies in its current form.

General comments :

The authors consistently emphasize "blockade/inhibition" of NR2F6 while they use genetic models (either KO-mice or siRNA in human cells). They do not have yet a blocking agent at hand, although the title and subtitles suggest this.

The only mechanistic insights how loss of NR2F6 enhances T cell activation are derived from the RNAseq study. Signaling should be confirmed and maybe blocking agents, genetic models etc. should be used to confirm the findings also on a functional level. This could be done using in vitro T cell assays.

Correlation of PD-1, PD-L1 and CTLA-4 expression with NR2F6 is interesting yet it cannot be concluded that this leads to synergistic effects of blocking antibodies

Tumor models are used very inconsistently (eg MCA-ind sarcoma and 5e4 MC38 in Fig 1 and 4, 7.5e5 MC38 in Fig 2, 5e5 MC38 in Fig 5, 5e5 B16 in S1 and 2, 1e5 B16 in Fig 4). Please comment and complete if necessary.

Myeloid population is totally ignored in this manuscript. Especially since aPDL1 is used, authors need to discuss the relative contribution of myeloid NR2F6 to the efficacy observed with or without aPDL1.

Specific comments :

The rationale for investigating NR2F6 is unclear. How did the authors come up with this target? This should be clarified.

Figure 1:

- Fig 1c/d: Are TILs mostly CD4 or CD8? Discriminate both populations within the CD3 pool
- Fig S1a-d:
 - o Which tumor model is used for these analyses? PD1 and PDL1 levels on CD4 and CD8 T cells are assessed by qPCR. FACS analysis would be more informative and confirm differences on a protein level.
 - o How are differences on PD1 and PDL1 expression when tumor suspensions are prepared and analysed by FACS directly ex vivo without stimulation? Differences only seem to come up after stimulation.
- The statement "improved CD8+ and CD4+ T cell effector functions in Nr2f6-/- mice" is not supported by data. No effector functions are reported in this figure.

- Having said that effector functions should be analysed not only in peripheral T cells (as per Fig 4), but also in CD4 and CD8 cells from TILs of B16 and MC38 tumors in WT and KO mice.
- Fig 1b: show tumor volume after d33 and survival would be informative to see if tumors are coming up later or not at all

Fig2:

- It remains unclear why KO mice are no longer able to induce significant tumor control when higher number of MC38/B16 cells are injected, as opposed to clear benefit observed when low cell numbers are injected. In-depth characterization of immune infiltration in those tumors should be performed.
- Are the CD4/CD8 cells (target population of this study) critical for the synergy observed? Depletion experiments should be performed in one of the tumor models.

Fig 3:

- The text within the figure is too small and hard to read. Same is true for supplementary figures and tables accompanying this figure.
- Please ensure Fig legend of Fig S3 has correct labeling of sub-panels

Fig 4:

- Were there any differences in overall survival of homozygous and heterozygote animals ?
- Did heterozygous animals respond to aPDL1 similar to homozygous? This is important to be shown since a comment about inefficient targeting of NR2F6 seems to be sufficient is made by the authors in the discussion.
- Again, intratumoral T cells have not been analyzed and should be analyzed.
- Regarding tumor volumes +/- mice show similar phenotype to -/- mice supporting the haplo-insufficient effect of Nr2f6. However, looking at immune infiltrates the results rather suggest significant differences. How can the authors explain this?

Fig 5:

- Fig 5a-d: Does adoptive transfer of Nr2f6 silenced T cells also lead to delay in tumor growth without PD-L1 blockade? It would be interesting to have all four groups as comparison (+/- anti-PD-L1 and +/- silencing).
- It is required to phenotypically and functionally characterize the tumor infiltrating adoptively transferred WT/siRNA T cells to provide a link between NR2F6 silencing and anti-tumor immune responses (PD1, PDL1, IL2, IFNg, Ki67). This would be a key finding which would show that adoptively transferred T cells (specifically lacking NR2F6) are efficient in effector functions.
- Fig 5 f and i: analysis of IL2 and IFNg levels on protein level by FACS or ELISA would be more informative than qPCR.

Fig 6:

- Authors state that no correlation between NR2F6 expression and O.S was observed, but patients with high score for NR2F6 (>3-4) appear to have improved OS. Please confirm and explain.
- Fig 6 D could potentially be shown graphically for comparisons of interest, with the detailed table included in the supplementary section.
- PDL1 expression on immune cells should be included and correlation to NR2F6 should be reported.
- Representative IHC images for all antigens reported in the table should be included in the suppl section.

Response to Reviewers' comments:

I would like to thank the referees for the constructive and indeed very helpful comments and suggestions regarding our manuscript entitled "Nuclear Receptor NR2F6 Inhibition Potentiates Responses to PD-L1/PD-1 Cancer Immune Checkpoint Blockade".

We believe that these additional data are particularly relevant and directly address the vast majority of the reviewer's comments. Particularly, changes in the revised MS addressing the major concerns about our initial submission in April 2017 are:

- (i) Confirming our immunohistochemistry (IHC) data of upregulated lymphatic NR2F6 expression in NSCLC patients with quantitative RT-PCR,
- (ii) Performing *in vitro* time kinetics investigating how long NR2F6 function is suppressed in T cells after *Nr2f6* siRNA knockdown,
- (iii) Analyses to anti-PD-L1 response of both homo- and heterozygous animals,
- (iv) Depletion effects of CD4 and/or CD8 cells on tumor growth of *Nr2f6* knockout animals,
- (v) Experiments to characterize the adoptively transferred *Nr2f6* siRNA T cells *in vivo*.

Below, please find our point-by-point response to all the questions/concerns/suggestions of the reviewers.

Taken together, the novel experimental data further strengthen our conclusion. These new results markedly support our initial observations and strongly substantiate the biological significance of effector T cell-intrinsic NR2F6 for the rejection of tumors. We also agree that the requested clarifications and revisions are relevant and have significantly improved the scientific quality of our findings. We hope that you will find that our revised paper now merits to be accepted for publication in Nature Communications.

Reviewer #1

In previous works, Baier and Hermann-Kleiter demonstrated that the nuclear orphan receptor NR2F6 is a PKC substrate and a suppressor of T cell activation and of Th17-dependent autoimmunity using a full germline knockout. Later they published a paper in Cell Reports (2015) demonstrating that a genetical deficiency in NR2F6 delays the growth of cancer cell in a model of prostate cancer and using the B16 melanoma as well. In the last paper, the authors demonstrated by adoptive transfer experiments that NR2F6 deficiency improves the ability of T cells to reject the tumor, indicating that the negative effect of NR2F6 is T cell-intrinsic. In the current paper, Baier and coworkers use a combination of genetic deficiency in NR2F6 with an anti-PDL1 antibody as a demonstration that the combination of an existing checkpoint inhibitor with a future treatment aimed to inhibit NR2F6 would make sense from the point of view of impacting on non-redundant inhibitory pathway.

We thank Reviewer #1 for his/her encouraging and insightful comments with regard to novelty and potential impact of our manuscript.

Reply to the remarks of Reviewer #1

The results are interesting but represent just some incremental value over their previous work in Cell Reports. A piece of information that could make the paper much more relevant could be the expansion of the data contained in Figure 6. In this Figure, the authors try to show that tumor infiltrating T cells in human lung cancer tumors show upregulated NR2F6 expression indicating that this orphan receptor could be an interesting target for future therapies in humans. However, the data of Figure 6 is supported by immunohistochemistry data, which in my opinion is not sufficiently quantitative to sustain the authors' claims. I would recommend that the authors try to generate more data in that direction, isolating TILs from human melanoma or other amenable tumors and using qPCR, intracellular flow cytometry or any other method that can provide real quantitative data.

Response to your point 1: We want to thank for his/her thoughtful advice to help further improve our manuscript. As the reviewer rightfully comments, the conclusion drawn in our previous Fig. 6 did rely on IHC data from a NSCLC patient cohort. Because an additional method that can provide real quantitative data would be indeed advantageous to confirm these IHC results, we devised experiments with our local clinical oncology department. Freshly collected NSCLC tumor samples (directly frozen after surgery) have been made available to us.

Initial attempts of intracellular FACS staining, however, turned out to be exceedingly difficult to perform with CD3⁺ T cells isolated from thawed tumor as well as from PBMC materials. Notwithstanding this experimental limitation, we FACS purified these CD3⁺ TIL populations derived from NSCLC biopsies to perform qRT-PCR. Thawed single cell suspensions from tumors and PBMC were stained with antibodies against tumor marker EpCam plus CD45 and CD3, and EpCam negative, CD45 and CD3 positive cells were sorted directly into the RNA lysis buffer on a FACSAria I.

As result and in support of our previous IHC data, we observed a significantly increased NR2F6 mRNA expression in CD3⁺ TILs from 3 out of the 5 NSCLC biopsies (see new **Figure 6C**) that we were able to obtain by our clinical partners. We agree that these quantitative RT-PCR results indeed reinforce our conclusion of an upregulated of NR2F6 mRNA expression in TILs in defined NSCLC patient samples. The text of the revised MS was changed accordingly.

Reviewer #2

The study of Klepsch et al. is build on prior work of the same group characterizing the impact of nuclear orphan receptor NR2F6 deficiency on antitumoral immunity. In prior studies the group had shown that genetic ablation of NR2F6 is associated with an improved tumor control in an autochtonus prostate cancer tumor model and in a syngeneic B16 melanoma model. The current study extends the previous work by investigating NR2F6 ablation in combination with PD-L1 blockade exploiting B16OVA and MC38 syngeneic tumor models. The key finding supported by a set of independent experiments is the synergistic effects of combined NR2F6 ablation PD-L1 blockade. The studies are well performed, findings are convincing and conclusions sound. The findings are highly relevant and might have a translational impact.

We want to thank Reviewer 2 for acknowledging the novelty of our findings. We also want to thank for his/her thoughtful advice to help further improve our manuscript.

Reply to the remarks of Reviewer #2

The study needs minor revision to be acceptable.

Comment 1): the group uses genetically modified mice that were backcrossed to C57/B6. The group should provide information how many rounds of backcrossing was performed. I did not find any information in their previous Cell Report publication, neither.

Response to your point 1: We apologize for this omission. *Nr2f6*-deficient mice have been backcrossed on C57BL/6 8-times and this information has now been added to the Materials and Methods section in our revised manuscript.

Comment 2) I am intrigued but also surprised that the adoptive transfer of transiently NRF2F6 siRNA transfected primary polyclonal CD3 is sufficient to improve tumor control in WT MC38 tumor bearing mice. I would like to see *in vitro* time kinetic data investigating how long (i) NR2F6 is suppressed in T cells after siRNA knockdown and (ii) this affects functional on *in vitro* T cell activation.

Response to your point 2: We agree with the statement of the reviewer and, for the revised version of our manuscript, we extended our *in vitro* analyses to confirm the induced hyper-responsiveness of murine T cell upon siRNA-mediated *Nr2f6* knockdown over a longer time period. When monitoring the cells over a period of up to 7 days after *Nr2f6* siRNA transfection, stimulation of *Nr2f6* silenced CD8⁺ T cells showed significantly elevated IFN γ responses detectable over the entire time period of 7 days. This data have now been added, as requested, as new **Supplementary Figure 5 G&F** to our revised manuscript. Thus, enhanced effector functions in *Nr2f6* siRNA transfected T cells remain stable over 7 days, which justifies the 7 day distance between the ACTs.

Comment 3.) Since the study is already extensive I do not request additional in vivo experiments. However, the authors should discuss potential mechanisms involved in the improved tumor control including priming and effector phase.

Response to your point 3: We thank Reviewer 2 for this insightful comment and have included a better statement of the potential underlying mechanisms leading to the observed superior cancer immune response associated with genetic *Nr2f6* inhibition (please see new paragraph in the discussion section): “Because lymphatic NR2F6 acts as a potent and selective repressor of effective cancer immunity, NR2F6 represents a crucial regulator of cancer immune tolerance induction and maintenance. Mechanistically, as examined by RNA sequencing of TILs derived from tumour-bearing mice, NR2F6 appears to set the threshold of T cell effector functions as transcriptional regulator of critical target genes shaping activation, recruitment, proliferation and homeostasis of tumour-antigen-specific T cell responses.”

Reviewer #3

The authors demonstrate that loss or reduction of the nuclear factor NR2F6 lowers the threshold for T cell activation in vitro in murine and human T cells. In mice lacking NR2F6, PD-1 blockade was more efficient in different s.c. tumor models. Also adoptive transfer of T cells with NR2F6 silencing reduced tumor growth, a finding that could improve current adoptive cell transfers. This manuscript reports data of significant value and provides meaningful information on NR2F6 as a candidate target for combinatorial immunotherapies. Yet, the conclusions made by the authors are not completely justified by the results they present. There are several weaknesses and inconsistencies in its current form.

We thank Reviewer #3 for the detailed and insightful comments on our manuscript and the potential impact of our study. We have carefully addressed basically all points of critique raised by the reviewer and included substantial amounts of new data in our revised manuscript. We believe that addressing these concerns significantly improved the quality of our manuscript, substantiating our conclusions. Please see below the detailed reply to his/her comments.

Reply to the remarks of Reviewer #3

General comments: The authors consistently emphasize “blockade/inhibition” of NR2F6 while they use genetic models (either KO-mice or siRNA in human cells). They do not have yet a blocking agent at hand, although the title and subtitles suggest this.

Response to your point 1: We thank the reviewer for this important remark. I have been remiss in not being clear in this point. This has been rephrased to genetic ablation or genetic inhibition throughout the revised version of the manuscript.

The only mechanistic insights how loss of NR2F6 enhances T cell activation are derived from the RNAseq study. Signaling should be confirmed and maybe blocking agents, genetic models etc. should be used to confirm the findings also on a functional level. This could be done using in vitro T cell assays.

Response to your point 2: Following the reviewer’s advice, we investigated the role of NR2F6 in lymphocyte activation and its metabolic process analysing extracellular acidification experimentally confirming one of our RNAseq findings (see new **Supplementary Figure 3C**).

Correlation of PD-1, PD-L1 and CTLA-4 expression with NR2F6 is interesting yet it cannot be concluded that this leads to synergistic effects of blocking antibodies

Response to your point 3: We were remiss in not addressing this purely speculative point correctly and we have rephrased our original statement in the revised version of the manuscript.

Tumor models are used very inconsistently (e.g. MCA-induced sarcoma and 5e4 MC38 in Fig 1 and 4, 7.5e5 MC38 in Fig 2, 5e5 MC38 in Fig 5, 5e5 B16 in S1 and 2, 1e5 B16 in Fig 4). Please comment and complete if necessary.

Response to your point 4: We apologize for the confusion in the initial version of our manuscript and now better define the model systems. We always used a high dosage of tumor cells (5e5) when applying a therapy like anti-PD-L1 or ACTs to ensure comparable tumor growth since we tested the originally used tumor load (1e5) with IgG2b alone and could see a rejection phenotype. Half of the B16-OVA cell number was used for MC38 because this cell line tends to be more aggressive in our hands without therapy. For the combined therapy (Fig5 siRNA ACT and anti-PD-L1 therapy) we increased the tumor load once more to really challenge our system and circumvent rejection. This information has now been added to our revised manuscript to improve clarity.

Myeloid population is totally ignored in this manuscript. Especially since aPDL1 is used, authors need to discuss the relative contribution of myeloid NR2F6 to the efficacy observed with or without aPDL1.

Response to your point 5: As the suggestion of the referee is valid we performed an analysis of the innate immune response. As result, no difference in the tumor-infiltrating myeloid cell populations (monocytes and macrophages) could be observed. In support of this view, no bone marrow egress phenotype was observed in *Nr2f6* knockout mice. Albeit our results certainly do not rule out any effects of *Nr2f6* deficiency on immune cells other than T cell, we conclude from our data, that *Nr2f6* deficiency does in fact not appear to directly regulate innate immune cells at the tumor site. The text in the manuscript was changed accordingly.

Specific comments: The rationale for investigating NR2F6 is unclear. How did the authors come up with this target? This should be clarified.

Response to your point 6: This is a good point of the referee and I am sorry, that I did not to make it clear in the beginning. I have been working on the characterization of T cell functions of the PKC family members in rodent models and human cells for 25 years, and during those years I obtained the identification of their essential downstream signalling pathways and direct effector substrates such as NR2F6 as key regulators of the induction and/or maintenance of peripheral immunological tolerance. This historic angle has been described in the Immunity 2008 paper cited in the revised manuscript as well as our recent reviews:

Klepsch V, Hermann-Kleiter N, Baier G. Beyond CTLA-4 and PD-1: Orphan nuclear receptor NR2F6 as T cell signalling switch and emerging target in cancer immunotherapy. *Immunol Lett.* 2016 Mar 15. pii: S0165-2478(16)30032-3.

Hermann-Kleiter N, Baier G. Orphan nuclear receptor NR2F6 acts as an essential gatekeeper of Th17 CD4⁺ T cell effector functions. *Cell Commun Signal.* 2014 June;12(1):38.

Figure 1: • Fig 1c/d: Are TILs mostly CD4 or CD8? Discriminate both populations within the CD3 pool

Response to your point 7: Following the reviewers suggestion, the results of these analyses are now included as new **Supplementary Figure 11** of the revised version of the manuscript. There are no apparent differences between CD4 and CD8 subsets in TILs in this MCA tumor model. This finding confirms our notion that NR2F6 similarly regulates CD4⁺ and CD8⁺ effector T cells.

• Fig S1a-d: Which tumor model is used for these analyses? PD1 and PDL1 levels on CD4 and CD8 T cells are assessed by qPCR. FACS analysis would be more informative and confirm differences on a protein level. How are differences on PD1 and PDL1 expression when tumor suspensions are prepared and analysed by FACS directly *ex vivo* without stimulation? Differences only seem to come up after stimulation.

Response to your point 8: The tumor model used is the B16-OVA model system and the infiltrate analysis thereof is shown in **Supplementary Fig.1 E&F**. We, however, apologize for an apparent misunderstanding as this experimental data set in **Supplementary Fig.1 A-D** has been generated from sorted T cells *in vitro*. The text in the revised legends was changed accordingly to improve clarity.

• The statement “improved CD8+ and CD4+ T cell effector functions in *Nr2f6*^{-/-} mice” is not supported by data. No effector functions are reported in this figure. • Having said that effector functions should be analysed not only in peripheral T cells (as per Fig 4), but also in CD4 and CD8 cells from TILS of B16 and MC38 tumors in WT and KO mice.

Response to your point 9: We thank the reviewer for identifying this over statement and it has been changed accordingly in the revised version of the manuscript.

• Fig 1b: show tumor volume after d33 and survival would be informative to see if tumors are coming up later or not at all

Response to your point 10: We were remiss in not addressing this important point. Following the reviewer’s valuable advice, we investigated the role of *Nr2f6*-deficiency on overall survival well beyond day 33 and have thus added this data to our revised manuscript (see new **Fig. 1C**). Of note, tumor volume did not increase - tumors were not coming at all in the *Nr2f6* knockout group (not shown).

Fig2: • It remains unclear why KO mice are no longer able to induce significant tumor control when higher number of MC38/B16 cells are injected, as opposed to clear benefit observed when low cell numbers are injected. In-depth characterization of immune infiltration in those tumors should be performed

Response to your point 11: We thank the reviewer for her/his valid comments. To our opinion, there still is a clear tumor growth inhibition benefit and it's merely a doses effect of these aggressive tumor cell lines used. Of note, however, immune infiltration analyses *in vivo* of the *Nr2f6* knockout group in those high-dose tumors models, in principle, recapitulated the situation of the low dose model (see **Supplementary Fig3A&B** and data not shown) with an increased CD3⁺ T cell number. A comment has been added to our revised manuscript to improve clarity.

- Are the CD4/CD8 cells (target population of this study) critical for the synergy observed? Depletion experiments should be performed in one of the tumor models

Response to your point 12: As the reviewer correctly points out, an adaptive immune response against tumor depend both on CD4 and CD8 effector cells. We thus treated *Nr2f6*-deficient mice and control cohorts with CD4 and/or CD8 depleting antibodies as an experiment to directly test their relative contribution in the observed tumor rejection phenotype of *Nr2f6*-knockout mice. As a result, the depletion of both subsets completely abolished the growth inhibition benefit of *Nr2f6*-deficient animals (see new **Supplementary Figure 2F**). The results of either CD4 or CD8 single depletion, albeit showing a trend, did not reach statistical significance. Consistent with the previous tumor infiltrate analyses (see e.g. our Cell Reports 2015 paper as REF#23) NR2F6 apparently regulates the amplitude of anti-cancer immunity by both the CD4⁺ and CD8⁺ T cell subsets.

Fig 3: • The text within the figure is too small and hard to read. Same is true for supplementary figures and tables accompanying this figure

Response to your point 13: We thank the reviewer for identifying this unreadable labelling in the initial version of our figure 3. We apologize for this mistake and it has been now enlarged accordingly in our **revised Fig. 3** as well as **revised Supplementary Fig.3**.

- Please ensure Fig legend of Fig S3 has correct labeling of sub-panels

Response to your point 14: We thank the reviewer for her/his valid comments. This Supplementary Fig. legend 3 D-F is correctly labelled in the revised version of the manuscript.

Fig 4: • Were there any differences in overall survival of homozygous and heterozygote animals • Did heterozygous animals respond to aPDL1 similar to homozygous? This is important to be shown since a comment about inefficient targeting of NR2F6 seems to be sufficient is made by the authors in the discussion. Again, intratumoral T cells have not been analyzed and should be analyzed.

Response to your point 15: As the suggestion of the referee is highly valid we added additional *in vivo* analysis of homozygous and heterozygote animals in our revised manuscript. As a clear result, we could neither detect any differences in overall survival (please see new **Fig.4B&D**) nor response outcome to anti-PD-L1 blockade (please see new **Fig.4E**) in the revised version of the manuscript, both key observations that indeed support our original statement. Following the reviewer's suggestion, the results of the infiltrate analyses of tumor-bearing heterozygote animals similarly demonstrate enhanced numbers of CD45⁺ cells. A comment has been added to our revised manuscript to improve clarity.

• Regarding tumor volumes +/- mice show similar phenotype to -/- mice supporting the haplo-insufficient effect of Nr2f6. However, looking at immune infiltrates the results rather suggest significant differences. How can the authors explain this?

Response to your point 16: We apologize for this apparent misunderstanding as this experimental data set shown in **Fig.4F&G** has been generated *in vitro*. The text in the revised legends was changed accordingly to improve clarity.

Fig 5: • Fig 5a-d: Does adoptive transfer of Nr2f6 silenced T cells also lead to delay in tumor growth without PD-L1 blockade? It would be interesting to have all four groups as comparison (+/- anti-PD-L1 and +/- silencing). • It is required to phenotypically and functionally characterize the tumor infiltrating adoptively transferred WT/siRNA T cells to provide a link between NR2F6 silencing and anti-tumor immune responses (PD1, PDL1, IL2, IFNg, Ki67). This would be a key finding which would show that adoptively transferred T cells (specifically lacking NR2F6) are efficient in effector functions.

Response to your point 17: We apologize for the incomplete information here. While Nr2f6 siRNA alone did also show an effect, it particularly acts as a robust “sensitizer” for the established anti-PD-L1 immune checkpoint blockade in mouse tumor models *in vivo*, correlating with significantly increased survival times (**Fig.5A-D** and data not shown). We have added this important statement in the revised version of the manuscript.

To address the comment of characterizing the adoptively transferred *Nr2f6* siRNA T cells *in vivo*, we have tested siRNA-silenced T cells in several ACT experiments. Albeit we can firmly demonstrate that *Nr2f6*-silenced T cells sensitize therapeutic anti-PD-L1 anti-tumor response in the B16-OVA model (results that are consistent with the observed benefit of *Nr2f6* mutant mice), your specific request of tracking polyclonal T cells, that have been *Nr2f6* siRNA transfected prior ACT in tumor bearing immune-competent congenic wild-type recipient mice turned out to not yield a meaningful outcome (see inset figure below).

Albeit analysis of IL-2-producing polyclonal T cells in the dLN show elevated IL-2 levels in *Nr2f6* siRNA transfected CD4⁺ cells in mice receiving PD-L1 blockade therapy (see panel D below), there have been simply too few adoptively transferred T cells at the tumor site (see panel E below). Because of this inclusive result, the data are not shown in the revised MS, but are shown below as inset figure in this reply letter. A comment has been added to our revised manuscript to improve clarity.

We hope that this is acceptable for you.

Inset figure: Competitive adoptive cell transfer of control or *Nr2f6* siRNA transfected polyclonal CD3⁺ cells into immune-competent wild-type mice:

(A) Schematic representation of experimental setup. CD45.1xCD45.2 heterozygous mice were inoculated s.c. with 5×10^5 B16-OVA cells and were either treated with anti-PD-L1 blockade or IgG2b. On days 3 and 10 of tumor challenge, mice received simultaneous adoptive cell transfers (ACT) of 1×10^7 CD3⁺ cells transfected with control (ctr) siRNA (CD45.1) or *Nr2f6* siRNA (CD45.2). (B) Tumor growth kinetics in mice treated with ACT + PD-L1 blockade or ACT + IgG2b (n=4) (C) Representative dot plots of CD45.1⁺ and CD45.2⁺ transferred CD3⁺ cells in the draining lymph node (dLN) or tumor of CD45.1.2 heterozygous recipients. (D) Analysis of IL-2- producing CD4⁺ and CD8⁺ cells in the dLN show elevated IL-2 levels in *Nr2f6* siRNA transfected CD4⁺ (CD45.2) cells in mice receiving PD-L1 blockade therapy compared to recipient CD4⁺ (CD45.1.2) cells (p=0.0172) and control siRNA (CD45.1) transfected cells (p=0.0319). (E) Representative dot plots of IFNγ and IL-2 production in CD8⁺ cells in the tumor. Percentage of positive cells relative to parental gate is shown. Error bars represent the mean ± SEM.

- Fig 5 f and i: analysis of IL2 and IFNg levels on protein level by FACS or ELISA would be more informative than qPCR

Response to your point 18: As the suggestion of the referee is valid we added BioPlex analysis of IL-2 and IFNgamma cytokine secretion responses (see new **Fig.5G**), confirming the quantitative RT-PCR results. The text in the revised manuscript was changed accordingly.

Fig 6: • Authors state that no correlation between NR2F6 expression and O.S was observed, but patients with high score for NR2F6 (>3-4) appear to have improved OS. Please confirm and explain.

Response to your point 19: We thank the reviewer for this remark. I have been remiss in not being clear in this important point. There is no significant correlation between lymphatic NR2F6 expression and O.S. in any of these groups. This information has now been added in our revised manuscript to improve clarity.

- Fig 6 D could potentially be shown graphically for comparisons of interest, with the detailed table included in the supplementary section.
- PDL1 expression on immune cells should be included and correlation to NR2F6 should be reported.
- Representative IHC images for all antigens reported in the table should be included in the suppl section.

Response to your point 20: In line with the request of the referee, we have added the representative IHC images for all antigens as **new Supplementary Fig.6**. However, due to the limited biopsy material, it was impossible for us to re-evaluate the NSCLC cohort for lymphatic PD-L1 expression. We hope that this is acceptable.

REVIEWERS' COMMENTS:

Reviewer #1 (Remarks to the Author):

The authors have satisfactorily answered my previous concerns

Reviewer #3 (Remarks to the Author):

The authors sufficiently addressed all points. I do not have any further concerns and would recommend to publish the manuscript.

Editorial Note: Reviewer #3 feels you have adequately addressed Reviewer#2 concerns.